# On the Stability of Iterative Retraining of Generative Models on their own Data

**Quentin Bertrand**[*]
Université de Montréal and Mila

**Avishek (Joey) Bose**
McGill and Mila

**Alexandre Duplessis**
ENS, PSL

**Marco Jiralerspong**
Université de Montréal and Mila

**Gauthier Gidel**[†]
Université de Montréal and Mila

## Abstract

Deep generative models have made tremendous progress in modeling complex data, often exhibiting generation quality that surpasses a typical human's ability to discern the authenticity of samples. Undeniably, a key driver of this success is enabled by the massive amounts of web-scale data consumed by these models. Due to these models' striking performance and ease of availability, the web will inevitably be increasingly populated with synthetic content. Such a fact directly implies that future iterations of generative models will be trained on both clean and artificially generated data from past models. In this paper, we develop a framework to rigorously study the impact of training generative models on mixed datasets—from classical training on real data to self-consuming generative models trained on purely synthetic data. We first prove the stability of iterative training under the condition that the initial generative models approximate the data distribution well enough and the proportion of clean training data (w.r.t. synthetic data) is large enough. We empirically validate our theory on both synthetic and natural images by iteratively training normalizing flows and state-of-the-art diffusion models on CIFAR10 and FFHQ.

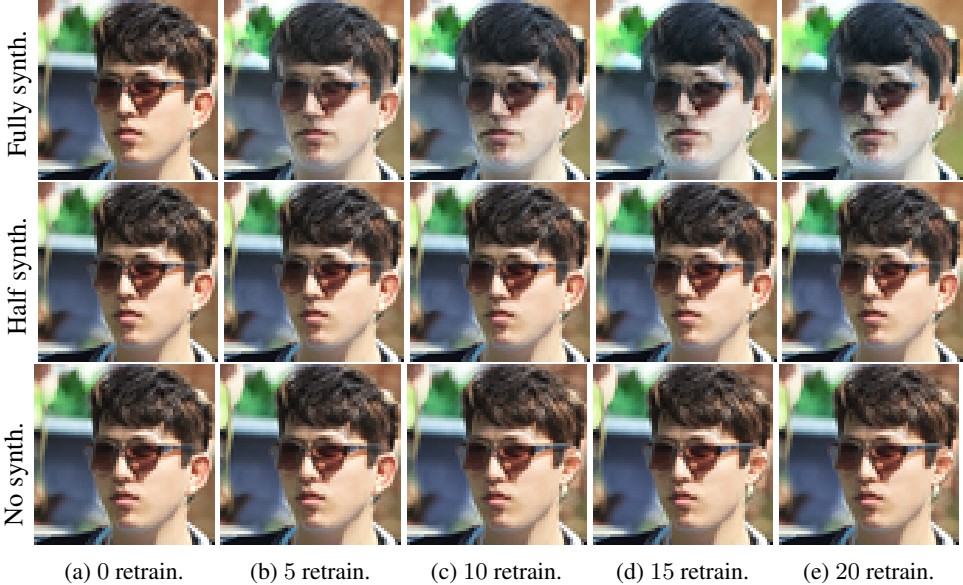

(a) 0 retrain.    (b) 5 retrain.    (c) 10 retrain.    (d) 15 retrain.    (e) 20 retrain.

Figure 1: **Samples generated from EDM trained on the FFHQ dataset.** As observed in Shumailov et al. (2023); Alemohammad et al. (2023), iteratively retraining the model exclusively on its own generated data yields degradation of the image (top row). On the other hand, retraining on a mix of half real and half synthetic data (middle) yields a similar quality as retraining on real data (bottom).

---

[*]Corresponding authors: `quentin.bertrand@mila.quebec`
[†]Canada Cifar AI Chair

# 1    INTRODUCTION

One of the central promises of generative modeling is the ability to generate unlimited synthetic data indistinguishable from the original data distribution. Access to such high-fidelity synthetic data has been one of the key ingredients powering numerous machine learning use cases such as data augmentation in supervised learning (Antreas et al., 2018), de novo protein design (Watson et al., 2023), and text-generated responses to prompts that enable a suite of natural language processing applications (Dong et al., 2022). Indeed, the scale and power of these models—in particular large language models (Brown et al., 2020; OpenAI, 2023; Chowdhery et al., 2022; Anil et al., 2023)—have even led to significant progress, often exceeding expert expectations (Steinhardt, 2022), in challenging domains like mathematical reasoning, or code assistance (Bubeck et al., 2023).

In addition to the dramatic increase in computational resources, a key driver of progress has been the amount and the availability of high-quality training data (Kaplan et al., 2020). As a result, current LLMs and text-conditioned diffusion models like DALL-E (Ramesh et al., 2021), Stable Diffusion (Stability AI, 2023), Midjourney (Midjourney, 2023) are trained on web-scale datasets that already potentially exhaust all the available clean data on the internet. Furthermore, a growing proportion of synthetic data from existing deep generative models continues to populate the web—to the point that even existing web-scale datasets are known to contain generated content (Schuhmann et al., 2022). While detecting generated content in public datasets is a challenging problem in its own right (Mitchell et al., 2023; Sadasivan et al., 2023), practitioners training future iterations of deep generative models must first contend with the reality that their training datasets *already* contain synthetic data. This raises the following fundamental question:

*Does training on mixed datasets of finite real data and self-generated data alter performance?*

**Our Contributions**. We study the iterative retraining of deep generative models on mixed datasets composed of clean real data and synthetic data generated from the current generative model itself. In particular, we propose a theoretical framework for iterative retraining of generative models that builds upon the standard maximum likelihood objective, extending it to retraining on mixed datasets. It encompasses many generative models of interest such as VAEs (Kingma and Welling, 2019), normalizing flows (Rezende and Mohamed, 2015), and state-of-the-art diffusion models (Song et al., 2021).

Using our theoretical framework, we show the stability of iterative retraining of deep generative models by proving the existence of a fixed point (Theorem 1) under the following conditions: 1.) The first iteration generative model is sufficiently "well-trained" and 2.) Each retraining iteration keeps a high enough proportion of the original clean data. Moreover, we provide theoretical (Proposition 1) and empirical (Figure 1) evidence that failure to satisfy these conditions can lead to model collapse (*i.e.,* iterative retraining leads the generative model to collapse to outputting a single point). We then prove in Theorem 2 that, with high probability, iterative retraining remains within a *neighborhood* of the optimal generative model in parameter space when working in the stable regime. Finally, we substantiate our theory on both synthetic datasets and high dimensional natural images on a broad category of models that include continuous normalizing flows (Chen et al., 2018) constructed using a conditional flow-matching objective (OTCFM, Tong et al. 2023), Denoising Diffusion Probabilistic Models (DDPM, Ho et al. 2020) and Elucidating Diffusion Models (EDM, Karras et al. 2022).

We summarize the main contributions of this paper below:

- We provide a theoretical framework to study the stability of the iterative retraining of likelihood-based generative models on mixed datasets containing self-generated and real data.
- Under mild regularity condition on the density of the considered generative models, we prove the stability of iterative retraining of generative models under the condition that the initial generative model is close enough to the real data distribution and that the proportion of real data is sufficiently large (Theorems 1 and 2) during each iterative retraining procedure.
- We empirically validate our theory through iterative retraining on CIFAR10 and FFHQ using powerful diffusion models in OTCFM, DDPM, and EDM.

# 2    ITERATIVE RETRAINING OF GENERATIVE MODELS

**Notation and convention**. Let $p_{\text{data}} \in \mathbb{P}(\mathbb{R}^d)$ be a probability distribution over $\mathbb{R}^d$. An empirical distribution sampled from $p_{\text{data}}$ in the form of a training dataset, $\mathcal{D}_{\text{real}} = \{\mathbf{x}_i\}_{i=1}^n$, is denoted $\hat{p}_{\text{data}}$.

---
**Algorithm 1** Iterative Retraining of Generative Models

---
**input** : $\mathcal{D}_{\text{real}} := \{\mathbf{x}_i\}_{i=1}^n, \mathcal{A}$ // True data, learning procedure (*e.g.*, Eq. 2)
**param:** $T, \lambda$ // Number of retraining iterations, proportion of gen. data
$p_{\boldsymbol{\theta}_0} = \mathcal{A}(\mathcal{D}_{\text{real}})$ // Learn generative model on true data
**for** $t$ *in* $1, \ldots, T$ **do**
  $\mathcal{D}_{\text{synth}} = \{\tilde{\mathbf{x}}_i\}_{i=1}^{\lfloor \lambda \cdot n \rfloor}$, with $\tilde{\mathbf{x}}_i \sim p_{\boldsymbol{\theta}_{t-1}}$ // Sample $\lfloor \lambda \cdot n \rfloor$ synthetic data points
  $p_{\boldsymbol{\theta}_t} = \mathcal{A}(\mathcal{D}_{\text{real}} \cup \mathcal{D}_{\text{synth}})$ // Learn generative model on synthetic and true data
**return** $p_{\boldsymbol{\theta}_T}$

---

We write $p_{\boldsymbol{\theta}}$ to indicate a generative model with parameters $\boldsymbol{\theta}$ while $\Theta$ is the entire parameter space. A synthetic dataset sampled from a generative model is written as $\mathcal{D}_{\text{synth}}$. The optimal set of parameters and its associated distribution are denoted as $\boldsymbol{\theta}^\star$ and $p_{\boldsymbol{\theta}^\star}$. We use subscripts, *e.g.*, $\boldsymbol{\theta}_t, \mathcal{D}_t$, to denote the number of iterative retraining steps, and superscripts to indicate the number of samples used, *e.g.*, $\boldsymbol{\theta}_t^n$ uses $n$ training samples at iteration $t$. Moreover, we use the convention that $\boldsymbol{\theta}_0$ is the initial generative model $p_{\boldsymbol{\theta}_0}$ trained on $\hat{p}_{\text{data}}$. The asymptotic convergence notation $u_t = \tilde{O}(\rho^t)$ where $\rho > 0$, refers to $\forall \epsilon > 0, \exists C > 0$ such that $u_t \leq C(\rho + \epsilon)^t, \forall t \geq 0$, *i.e.*, $u_t$ almost converges at a linear rate $\rho^t$. The Euclidean norm of vectors and the spectral norm of matrices is denoted $\|\|_2$. Let $\mathcal{S}_+$ be the set of symmetric positive definite matrices, for $S_1$ and $S_2 \in \mathcal{S}_+$, $S_1 \succeq S_2$ if $S_1 - S_2 \succeq 0$.

## 2.1 PRELIMINARIES

The goal of generative modeling is to approximate $p_{\text{data}}$ using a parametric model $p_{\boldsymbol{\theta}}$. For likelihood-based models, this corresponds to maximizing the likelihood of $\hat{p}_{\text{data}}$, which is an empirical sampling of $n$ data points from $p_{\text{data}}$ and leads to the following optimization objective:

$$\Theta_0^n := \underset{\boldsymbol{\theta}' \in \Theta}{\operatorname{argmax}} \, \mathbb{E}_{\mathbf{x} \sim \hat{p}_{\text{data}}}[\log p_{\boldsymbol{\theta}'}(\mathbf{x})] \ . \tag{1}$$

Note that the $\operatorname{argmax}$ operator defines a set of equally good solutions to Equation (1). We follow the convention of selecting $\boldsymbol{\theta}_0^n = \operatorname{argmin}_{\boldsymbol{\theta}' \in \Theta_0} \|\boldsymbol{\theta}'\|$ as the canonical choice. After obtaining the trained generative model $p_{\boldsymbol{\theta}_0}$, we are free to sample from it and create a synthetic dataset $\mathcal{D}_{\text{synth}} = \{\tilde{\mathbf{x}}_i^0\}_{i=1}^{\lfloor \lambda \cdot n \rfloor}$, where $\tilde{\mathbf{x}}_i^0 \sim p_{\boldsymbol{\theta}_0}$ and $\lambda > 0$ is a hyperparameter that controls the relative size of $\mathcal{D}_{\text{synth}}$ with respect to $\mathcal{D}_{\text{real}}$. We can then update our initial generative model on any combination of real data $\hat{p}_{\text{data}}$ and synthetic data drawn from $p_{\boldsymbol{\theta}_0}$ to obtain $p_{\boldsymbol{\theta}_1}$ and repeat this process ad infinitum. We term this procedure *iterative retraining* of a generative model (Algorithm 1). In the context of maximum likelihood, the *iterative retraining* update can be formally written as:

$$\Theta_{t+1}^n := \underset{\boldsymbol{\theta}' \in \Theta}{\operatorname{local-argmax}} \left[ \mathbb{E}_{\mathbf{x} \sim \hat{p}_{\text{data}}}[\log p_{\boldsymbol{\theta}'}(\mathbf{x})] + \lambda \mathbb{E}_{\tilde{\mathbf{x}} \sim \hat{p}_{\boldsymbol{\theta}_t}}[\log p_{\boldsymbol{\theta}'}(\tilde{\mathbf{x}})] \right] . \tag{2}$$

The notation $\operatorname{local-argmax}$ corresponds to any local maximizer of the loss considered.[1] The formulation Equation (2) precisely corresponds to training a model on the empirical dataset $\mathcal{D}_{\text{real}} \bigcup \mathcal{D}_{\text{synth}}$ of size $\lfloor n + \lambda \cdot n \rfloor$. Note that $\Theta_{t+1}^n$ may be a set; we thus need to provide a tie-breaking rule to pick $\boldsymbol{\theta}_{t+1}^n \in \Theta_{t+1}^n$. Since most training methods are *local search method*, we will select the *closest* solution to the previous iterate $\boldsymbol{\theta}_t^n$,[2]

$$\boldsymbol{\theta}_{t+1}^n = \mathcal{G}_\lambda^n(\boldsymbol{\theta}_t^n) := \underset{\boldsymbol{\theta}' \in \Theta_{t+1}^n}{\operatorname{argmin}} \|\boldsymbol{\theta}' - \boldsymbol{\theta}_t^n\| \ . \tag{3}$$

Setting $\lambda = 0$ corresponds to only sampling from $\hat{p}_{\text{data}}$ and is thus equivalent to training $p_{\boldsymbol{\theta}_0}$ for more iterations. The main focus of this work is studying the setting where $\lambda > 0$ and, more specifically, analyzing the discrete sequence of learned parameters $\boldsymbol{\theta}_0^n \to \boldsymbol{\theta}_1^n \to \cdots \to \boldsymbol{\theta}_T^n$ and thus their induced distributions and characterize the behavior of the generative model in reference to $p_{\text{data}}$. As a result, this means that iterative retraining at timestep $t + 1$—*i.e.*, $\boldsymbol{\theta}_{t+1}^n$—resumes from the previous iterate $\boldsymbol{\theta}_t^n$. In practice, this amounts to *finetuning* the model rather than retraining from scratch.

---
[1]Given a function $f$ continuously twice differentiable, a sufficient condition for $\boldsymbol{\theta}$ to locally maximize $f$ is $\nabla f(\boldsymbol{\theta}) = 0$ and $\nabla^2 f(\boldsymbol{\theta}) \prec 0$.

[2]For simplicity of presentation, we assume that such a $\boldsymbol{\theta}_{t+1}^n$ is uniquely defined which will be the case under the assumptions of Theorems 1 and 2.

## 2.2 Warm-up with Multivariate Gaussian

As a warm-up, we consider a multivariate Gaussian with parameters $\mathcal{N}_{\boldsymbol{\mu},\boldsymbol{\Sigma}}$ obtained from being fit on $\hat{p}_{\text{data}} \sim \mathcal{N}_{\boldsymbol{\mu}_0,\boldsymbol{\Sigma}_0}$. This example corresponds to a special case of Equation (2), where $\lambda \to \infty$ (*i.e.,* we do not reuse any training data). We analyze the evolution of the model parameters when iteratively retraining on its own samples. For each retraining, a multivariate Gaussian model is learned from finite samples. In the case of *a single* multivariate Gaussian, the unbiased update of the mean and covariance parameters has a closed-form formula. For all $t \geq 0$:

$$\text{Sampling step:} \quad \left\{ \mathbf{x}_t^j = \boldsymbol{\mu}_t + \sqrt{\boldsymbol{\Sigma}_t}\mathbf{z}_t^j, \text{ with } \mathbf{z}_t^j \overset{\text{i.i.d.}}{\sim} \mathcal{N}_{0,\mathbf{I}}, \; 1 \leq j \leq n \right. , \tag{4}$$

$$\text{Learning step:} \quad \left\{ \boldsymbol{\mu}_{t+1} = \frac{1}{n}\sum_j \mathbf{x}_t^j \; , \; \boldsymbol{\Sigma}_{t+1} = \frac{1}{n-1}\sum_j \left(\mathbf{x}_t^j - \boldsymbol{\mu}_{t+1}\right)\left(\mathbf{x}_t^j - \boldsymbol{\mu}_{t+1}\right)^\top \right. . \tag{5}$$

Proposition 1 states that retraining a multivariate Gaussian only on its generated data leads to its collapse: its covariance matrix vanishes to $0$ linearly as a function of the number of retraining.

> **Proposition 1.** *(Gaussian Collapse) For all initializations of the mean $\mu_0$ and the covariance $\Sigma_0$, iteratively learning a single multivariate Gaussian solely on its generated data yields model collapse. More precisely, if $\mu_t$ and $\Sigma_t$ follows Equations (4) and (5), then, there exists $\alpha < 1$,*
>
> $$\mathbb{E}(\sqrt{\boldsymbol{\Sigma}_t}) \preceq \alpha^t \sqrt{\boldsymbol{\Sigma}_0} \underset{t \to +\infty}{\longrightarrow} 0 \; . \tag{6}$$

The proof of Proposition 1 is provided in Appendix A. Interestingly, the convergence rate $\alpha_n$ goes to $1$ as the number of samples $n$ goes to infinity. In other words, the larger the number of samples, the slower the model collapses. Proposition 1 is a generalization of Shumailov et al. (2023); Alemohammad et al. (2023), who respectively proved convergence to $0$ of the standard deviation of a one-dimensional Gaussian model, and the covariance of a single multivariate Gaussian, *without rates of convergence*. While Proposition 1 states that learning a simple generative model retrained *only* on its own output yields model collapse, in Section 3, we show that if a sufficient proportion of real data is used for retraining the generative models, then Algorithm 1 is stable.

## 3 Stability of Iterative retraining

We seek to characterize the stability behavior of iterative retraining of generative models on datasets that contain a mixture of original clean data as well as self-generated synthetic samples. As deep generative models are parametric models, their performance is primarily affected by two sources of error: 1.) statistical approximation error and 2.) function approximation error. The first source of error occurs due to the sampling bias of using $\hat{p}_{\text{data}}$ rather than $p_{\text{data}}$. For instance, due to finite sampling, $\hat{p}_{\text{data}}$ might not contain all modes of the true distribution $p_{\text{data}}$, leading to an imperfect generative model irrespective of its expressive power. The function approximation error chiefly arises due to a mismatch of the model class achievable by $\boldsymbol{\theta}$, and consequently the induced distribution $p_{\boldsymbol{\theta}}$ and $p_{\text{data}}$.

Recent advances in generative modeling theory have proven the universal approximation abilities of popular model classes, *e.g.,* normalizing flows (Teshima et al., 2020) and diffusion models (Oko et al., 2023). In practice, state-of-the-art generative models such as StableDiffusion (Stability AI, 2023) and MidJourney (Midjourney, 2023) exhibit impressive qualitative generative ability which dampens the practical concern of the magnitude of the function approximation error in generative models. Consequently, we structure our investigation by first examining the stability of iterative training in the absence of *statistical error* in Section 3.1, which is an idealized setting. In Section 3.2, we study the stability of iterative retraining of practical generative models where the statistical error is non-zero, which is reflective of the setting and use cases of actual SOTA generative models in the wild.

## 3.1 Iterative retraining under no Statistical Error

We first study the behavior of iterative retraining under the setting of *no statistical error*. As the primary source of statistical error comes from using $\hat{p}_{\text{data}}$ we may simply assume full access to $p_{\text{data}}$

instead and an infinite sampling budget. In this setting, we define the optimal generative model $p_{\boldsymbol{\theta}^\star}$ with parameters $\boldsymbol{\theta}^\star$ as the solution to the following optimization problem,

$$\boldsymbol{\theta}^\star \in \underset{\boldsymbol{\theta}' \in \boldsymbol{\theta}}{\operatorname{argmax}} \, \mathbb{E}_{\mathbf{x} \sim p_{\text{data}}}[\log p_{\boldsymbol{\theta}'}(\mathbf{x})] \ . \tag{7}$$

Compared to Equation (1), infinitely many samples are drawn from $p_{\text{data}}$ in Equation (7); hence, there is no statistical error due to finite sampling of the dataset. As the capacity of the model class $(p_{\boldsymbol{\theta}})_{\boldsymbol{\theta} \in \boldsymbol{\theta}}$ is limited, the maximum likelihood estimator $p_{\boldsymbol{\theta}^\star}$ does not exactly correspond to the data distribution $p_{\text{data}}$. We note $\varepsilon$ is the Wasserstein distance (Villani et al., 2009) between $p_{\boldsymbol{\theta}^\star}$ and $p_{\text{data}}$,

$$\varepsilon = d_W(p_{\boldsymbol{\theta}^\star}, p_{\text{data}}) := \sup_{f \in \{\text{Lip}(f) \le 1\}} \mathbb{E}_{\mathbf{x} \sim p_{\boldsymbol{\theta}^\star}}[f(\mathbf{x})] - \mathbb{E}_{\mathbf{x} \sim p_{\text{data}}}[f(\mathbf{x})] \ , \tag{8}$$

where $\{\text{Lip}(f) \le 1\}$ is the set of 1-Lipschitz functions. We assume the Hessian of the maximum log-likelihood from Equation (7) has some regularity and $\boldsymbol{\theta}^\star$ is a strict local maximum.

> **Assumption 1.** *For $\boldsymbol{\theta}$ close enough to $\boldsymbol{\theta}^\star$, the mapping $x \mapsto \nabla_{\boldsymbol{\theta}}^2 \log p_{\boldsymbol{\theta}}(\mathbf{x})$ is L-Lipschitz.*

We use this assumption to show that if $p_{\boldsymbol{\theta}^\star}$ is close to $p_{\text{data}}$ (*i.e.*, $\varepsilon$ small), then the Hessians of the maximum likelihood with respectively real and synthetic data are close to each other.

> **Assumption 2.** *The mapping $\boldsymbol{\theta} \mapsto \mathbb{E}_{\mathbf{x} \sim p_{\text{data}}}[\log p_{\boldsymbol{\theta}}(\mathbf{x})]$ is continuously twice differentiable locally around $\boldsymbol{\theta}^\star$ and $\mathbb{E}_{\mathbf{x} \sim p_{\text{data}}}[\nabla_{\boldsymbol{\theta}}^2 \log p_{\boldsymbol{\theta}^\star}(\mathbf{x})] \preceq -\alpha \mathbf{I}_d \prec 0$.*

Assumption 2 imposes local strong concavity around the local maximum $\boldsymbol{\theta}^\star$, and Assumption 1 implies Hessian Lipschitzness when close to $\boldsymbol{\theta}^\star$. For example, Assumptions 1 and 2 are valid for the Gaussian generative model described in Section 2.2.

> **Proposition 2.** *Let $p_{\boldsymbol{\theta}} : \mathbf{x} \mapsto \mathcal{N}_{\boldsymbol{\mu}, \boldsymbol{\Sigma}}(\mathbf{x}) = e^{-\frac{1}{2}(\mathbf{x}-\boldsymbol{\mu})^\top \boldsymbol{\Sigma}^{-1}(\mathbf{x}-\boldsymbol{\mu})} / \sqrt{(2\pi)^d |\boldsymbol{\Sigma}|}$, with $\boldsymbol{\mu} \in \mathbb{R}^d$, $\boldsymbol{\Sigma} \succ 0$, $\boldsymbol{\theta} = (\boldsymbol{\mu}, \boldsymbol{\Sigma}^{-1})$, then $\mathbb{E}_{\mathbf{x} \sim p_{\boldsymbol{\theta}}(\mathbf{x})}[\nabla_{\boldsymbol{\theta}}^2 \log p_{\boldsymbol{\theta}}(\mathbf{x})] \prec 0$ and $\mathbf{x} \mapsto \nabla_{\boldsymbol{\theta}}^2 \log p_{\boldsymbol{\theta}}(\mathbf{x})$ is 1-Lipschitz.*

A direct consequence of the proposition is that the class of generative models $(\mathcal{N}_{\boldsymbol{\mu}, \boldsymbol{\Sigma}})$ follows Assumptions 1 and 2. Thus, it proves that under these assumptions, a finite $\lambda$ is necessary for the stability of iterative retraining. Now, we study what conditions are sufficient for the stability of iterative retraining with $\lambda > 0$, *i.e.*, training by sampling from $\frac{1}{1+\lambda} p_{\text{data}} + \frac{\lambda}{1+\lambda} p_{\boldsymbol{\theta}^\star}$, which corresponds to,

$$\mathcal{G}_\lambda^\infty(\boldsymbol{\theta}) \in \underset{\boldsymbol{\theta}' \in \boldsymbol{\theta}}{\operatorname{local-argmax}} (\underbrace{\mathbb{E}_{\mathbf{x} \sim p_{\text{data}}}[\log p_{\boldsymbol{\theta}'}(\mathbf{x})]}_{:=\mathcal{H}_1(\boldsymbol{\theta}')} + \lambda \underbrace{\mathbb{E}_{\tilde{\mathbf{x}} \sim p_{\boldsymbol{\theta}}}[\log p_{\boldsymbol{\theta}'}(\tilde{\mathbf{x}})]}_{:=\mathcal{H}_2(\boldsymbol{\theta}, \boldsymbol{\theta}')}) \ . \tag{9}$$

If ties between the maximizers are broken as in Equation (3),[3] we can show that the solution of Equation (9) is locally unique, ensuring that $\mathcal{G}_\lambda^\infty(\boldsymbol{\theta})$ is well defined and differentiable.

> **Proposition 3.** *(The Local Maximum Likelihood Solution is Unique) Let $\boldsymbol{\theta}^\star$ be a solution of Equation (7) that satisfies Assumptions 1 and 2. If $\epsilon L < \alpha$, then for all $\lambda > 0$ and $\boldsymbol{\theta}$ in a small enough neighborhood $\mathcal{U}$ around $\boldsymbol{\theta}^\star$, there exists a unique local maximizer $\mathcal{G}_\lambda^\infty(\boldsymbol{\theta})$ in $\mathcal{U}$.*

The proof of Proposition 3 is provided in Appendix C. Interestingly, the solution $\boldsymbol{\theta}^\star$ of the maximum infinite sample likelihood Equation (7) is a fixed point of the infinite sample iterative retraining procedure via maximum likelihood in Equation (9).

---

[3]Formally, $\mathcal{G}_\lambda^\infty(\boldsymbol{\theta}) = \operatorname{argmin}_{\boldsymbol{\theta}' \in \bar{\boldsymbol{\theta}}} \|\boldsymbol{\theta}' - \boldsymbol{\theta}\|$, $\bar{\boldsymbol{\theta}} = \operatorname{local-argmax}_{\boldsymbol{\theta}' \in \boldsymbol{\theta}} \mathcal{H}_\lambda(\boldsymbol{\theta}, \boldsymbol{\theta}')$.

**Proposition 4.** *(The Optimal Parametric Generative Model is a Fixed Point) For a given data distribution $p_{\text{data}}$, any $\boldsymbol{\theta}^\star$ solution of Equation (7), and for all $\lambda > 0$ we have,*

$$\boldsymbol{\theta}^\star = \mathcal{G}_\lambda^\infty(\boldsymbol{\theta}^\star) \ . \tag{10}$$

The proof of Proposition 4 can be found in Appendix D. In the zero statistical error case, $\boldsymbol{\theta}^\star$ corresponds to the perfectly trained generative model obtained after the first iteration of training exclusively on real data (see Equation (1)). The main question is the stability of iterative retraining around $\boldsymbol{\theta}^\star$. Even with $p_{\text{data}}$, one can consider small sources of error such as optimization error or numerical error in floating point precision leading to an initial $\boldsymbol{\theta}_0 \approx \boldsymbol{\theta}^\star$ only approximately solving Equation (7).

For our stability analysis, given $\boldsymbol{\theta}$, we decompose the maximum likelihood loss as $\mathcal{H}_\lambda(\boldsymbol{\theta}, \boldsymbol{\theta}') := \mathcal{H}_1(\boldsymbol{\theta}') + \lambda\mathcal{H}_2(\boldsymbol{\theta}, \boldsymbol{\theta}') := \mathbb{E}_{\tilde{x}\sim p_{\text{data}}}[\log p_{\boldsymbol{\theta}'}(\tilde{\mathbf{x}})] + \lambda\mathbb{E}_{\mathbf{x}\sim p_{\boldsymbol{\theta}}}[\log p_{\boldsymbol{\theta}'}(\mathbf{x})]$. With this decoupling, we can view $\mathcal{H}_2$ as a regularizer injecting synthetic data and $\lambda$ as the amount of regularization. Our first main result states that iterative retraining of generative models is asymptotically stable if the amount of real data is "large enough" with respect to the amount of synthetic data.

**Theorem 1.** *(Stability of Iterative Retraining) Given $\boldsymbol{\theta}^\star$ as defined in Equation (7) that follows Assumptions 1 and 2, we have that, if $\lambda(1 + \frac{L\varepsilon}{\alpha}) < 1/2$, then the operator norm of the Jacobian is strictly bounded by 1, more precisely,*

$$\|\mathcal{J}_{\boldsymbol{\theta}}\mathcal{G}_\lambda^\infty(\boldsymbol{\theta}^\star)\|_2 \leq \frac{\lambda(\alpha + \varepsilon L)}{\alpha - \lambda(\alpha + \varepsilon L)} < 1 \ . \tag{11}$$

*Consequently, there exists a $R > 0$ such that for all $\boldsymbol{\theta}_0 \in \boldsymbol{\theta}$ that satisfy $\|\boldsymbol{\theta}_0 - \boldsymbol{\theta}^\star\| \leq R$, then starting at $\boldsymbol{\theta}_0$ and having $\boldsymbol{\theta}_{t+1} = \mathcal{G}_\lambda^\infty(\boldsymbol{\theta}_t)$ we have that $\boldsymbol{\theta}_t \underset{t\to+\infty}{\longrightarrow} \boldsymbol{\theta}^\star$ and*

$$\|\boldsymbol{\theta}_t - \boldsymbol{\theta}^\star\| = \tilde{O}\left(\left(\frac{\lambda(\alpha + \varepsilon L)}{\alpha - \lambda(\alpha + \varepsilon L)}\right)^t \|\boldsymbol{\theta}_0 - \boldsymbol{\theta}^\star\|\right) \ . \tag{12}$$

Proof of Theorem 1 can be found in Appendix E. Theorem 1 establishes the stability property of generative models' iterative retraining and quantifies the convergence rate to the fixed point $\boldsymbol{\theta}^\star$. Interestingly, setting $\lambda$ small enough is always sufficient to get $\|\mathcal{J}_{\boldsymbol{\theta}}\mathcal{G}_\lambda^\infty(\boldsymbol{\theta}^\star)\| < 1$. However, it is important to notice that, to prove that the local maximizer of $\boldsymbol{\theta}' \mapsto \mathcal{H}(\boldsymbol{\theta}, \boldsymbol{\theta}')$ is unique, we needed to assume that $\varepsilon L < \alpha$. Similarly, we speculate that with additional assumptions on the regularity of $\mathcal{H}$, the neighborhood size $\delta$ for the local convergence result of Theorem 1 could be controlled with $\varepsilon$.

The limitation of Theorem 1 is that it is based on using infinite samples from $p_{\text{data}}$ and only provides a *sufficient* condition for stability. Informally, we show that if $d_W(p_{\boldsymbol{\theta}^\star}, p_{\text{data}})$ and $\lambda$ are small enough–*i.e.,* our class of generative model can approximate well enough the data distribution and we re-use enough real data–, then, the procedure of iterative training is stable. On the one hand, It remains an open question to show that similar conditions as the ones of Theorem 1 are *necessary* for stability. On the other hand, Proposition 2 implies that in the general case under Assumptions 1 and 2, a finite value for $\lambda$ is *necessary* since for a multivariate Gaussian, regardless of the value of $d_W(p_{\boldsymbol{\theta}^\star}, p_{\text{data}})$, if $\lambda$ is infinite (*i.e.,* we only use generated data) then we do not have stability and collapse to a generative model with a vanishing variance.

Our theoretical results point to a different conclusion than the one experimentally obtained by Alemohammad et al. (2023, Fig.3). In particular, stability may be ensured without injecting more 'fresh data'. Instead, a sufficient condition for stability is to have a good enough initial model ($\varepsilon$ small) and a sufficient fraction of 'fixed real data' ($\lambda$ small enough) for each iterative training step.

### 3.2 ITERATIVE RETRAINING UNDER STATISTICAL APPROXIMATION ERROR

We now turn our attention to iterative retraining for generative models under finite sampling. Motivated by generalization bounds in deep generative models (Yang and E, 2021a;b; Ji et al., 2021; Jakubovitz et al., 2019), we make the following additional assumption on the approximation capability

of generative models as a function of dataset size: we suppose we have a generalization bound for our class of models with a vanishing term as the sample size increases.

> **Assumption 3.** *There exists $a, b, \varepsilon_{\text{OPT}} \geq 0$ and a neighborhood $U$ of $\boldsymbol{\theta}^\star$ such that with probability $1 - \delta$ over the samplings, we have[a]*
>
> $$\forall \boldsymbol{\theta} \in U, \forall n \in \mathbb{N}, \|\mathcal{G}_\lambda^n(\boldsymbol{\theta}) - \mathcal{G}_\lambda^\infty(\boldsymbol{\theta})\| \leq \varepsilon_{\text{OPT}} + \frac{a}{\sqrt{n}} \sqrt{\log \frac{b}{\delta}} \ . \tag{13}$$
>
> ---
> [a]We could have kept a more general bound as $C(n, \delta)$, and this choice comes without loss of generality.

The constant $\varepsilon_{\text{OPT}}$ is usually negligible and mainly depends on the (controllable) optimization error. As for the term $a/\sqrt{n} \cdot \sqrt{\log b/\delta}$, it vanishes to $0$ when increasing $n$. Intuitively, this assumption means that, in the neighborhood of $\boldsymbol{\theta}^\star$, the practical iterate $\mathcal{G}_\lambda^n(\boldsymbol{\theta})$ can get as close as desired to the ideal parameter $\mathcal{G}_\lambda^\infty(\boldsymbol{\theta})$ by increasing the size of the sample from the dataset $\mathcal{D}_t$ used for retraining at iteration $t$ and thus decreasing the optimization error. On the one hand, it is relatively strong to assume that one can approximate $\mathcal{G}_\lambda^\infty(\boldsymbol{\theta})$ in the parameter space (usually such bounds regard the expected loss). On the other hand, the fact that this assumption is local makes it slightly weaker.

We now state our main result with finite sample size, with high probability, that iterative retraining remains within a neighborhood of the fixed point $\boldsymbol{\theta}^\star$.

> **Theorem 2.** *(Approximate Stability) Under the same assumptions of Theorem 1 and Assumption 3, we have that there exists $0 < \rho < 1$ and $R > 0$ such that if $\|\boldsymbol{\theta}_0^n - \boldsymbol{\theta}^\star\| \leq R$ then, with probability $1 - \delta$,*
>
> $$\|\boldsymbol{\theta}_t^n - \boldsymbol{\theta}^\star\| \leq \left( \varepsilon_{\text{OPT}} + \frac{a}{\sqrt{n}} \sqrt{\log \frac{bt}{\delta}} \right) \sum_{l=0}^t \rho^l + \rho^t \|\boldsymbol{\theta}_0 - \boldsymbol{\theta}^\star\|, \quad \forall t \in \mathbb{N}^* \ . \tag{14}$$

Proof can be found in Appendix F. The main takeaway on Theorem 2 is that the error between the iteratively retrained parameters $\boldsymbol{\theta}_t^n$ and $\boldsymbol{\theta}^\star$ can be decomposed in three main terms: the optimization error $\varepsilon_{\text{OPT}} \cdot \sum_{l=0}^t \rho^l$, the statistical error $a/\sqrt{n} \cdot \sqrt{\log bt/\delta} \cdot \sum_{l=0}^t \rho^l$, and the iterative retraining error $\rho^t \|\boldsymbol{\theta}_0 - \boldsymbol{\theta}^\star\|$. Interestingly, the error between $\boldsymbol{\theta}_t^n$ and $\boldsymbol{\theta}^\star$ can be controlled in the regime where the $\log$ of the number of retraining is much smaller than the number of samples, *i.e.,* in the regime $\log t \ll n$.

## 4 RELATED WORK

**Generative Models trained on Synthetic Data**. The study of generative models within a feedback loop has been a focus of recent contemporary works by Alemohammad et al. (2023); Shumailov et al. (2023). Both these works exhibit the model collapse phenomenon in settings where the generative model is iteratively retrained from scratch (Alemohammad et al., 2023) and finetuned (Shumailov et al., 2023). Similar in spirit to our work, Alemohammad et al. (2023) advocates for the injection of real data—and even to add more fresh data—to reduce model collapse but falls short of providing theoretical stability guarantees which we provide. The effect of generative models trained on web-scale datasets contaminated with synthetic samples has also been investigated; corroborating the degradation of generated sample quality (Martínez et al., 2023a;b; Hataya et al., 2023).

**Performative Prediction**. The nature of the stability results in Theorem 1 and Theorem 2 bear connections with the literature on "performative prediction" in supervised classification (Perdomo et al., 2020). In both cases, the outputs of the models affect the sampling distribution, which influences the objectives themselves. As such, we seek to characterize the existence and nature of fixed points. However, we note that performative prediction focuses on *supervised learning*, and the required assumption in iterative stability for generative models is weaker than those to ensure global convergence in the performative prediction literature. For instance, Perdomo et al. (2021) require strong convexity with respect to $\boldsymbol{\theta}$, while Mofakhami et al. (2023) need stronger continuity properties of the mapping $\boldsymbol{\theta} \mapsto p_{\boldsymbol{\theta}}$. In contrast, we only require assumptions on $p_{\boldsymbol{\theta}}$ around an optimal point $\boldsymbol{\theta}^\star$.

## 5 EXPERIMENTS

We now investigate the iterative retraining of deep generative models in practical settings[4]. As our real-world models ingest datasets sampled from empirical distributions, they necessarily operate under statistical error arising from finite sampling. Other sources of error that inhibit learning the perfect generative model—irrespective of the expressiveness of the architecture—include optimization errors from the learning process and other numerical errors from lack of numerical precision. Our goal with this empirical investigation is to characterize the impact on the performance of these models as a function of $\lambda$ and iterative retraining iterations while holding the size of the real dataset constant.

**Datasets and Models**. We perform experiments on synthetic toy data as found in Grathwohl et al. (2018), CIFAR-10 (Krizhevsky and Hinton, 2009), and FlickrFacesHQ $64 \times 64$ (FFHQ-64 Karras et al. 2019) datasets. For deep generative models, we conduct experiments with continuous normalizing flows (Chen et al., 2018) constructed using a conditional flow-matching loss (CFM Lipman et al. 2022; Tong et al. 2023) and two powerful diffusion models in Denoising Diffusion Probabilistic Models (DDPM, Ho et al. 2020) and Elucidating Diffusion Models (EDM, Karras et al. 2022) where we relied on the original codebases `torch-cfm` (`https://github.com/atong01/conditional-flow-matching`), ddpm-torch (`https://github.com/tqch/ddpm-torch`) and edm (`https://github.com/NVlabs/edm`) for faithful implementations.

**Results on Synthetic Data**. In Figures 3 and 4 in Appendix G we illustrate the learned densities of DDPM (Figure 3) on the *8 Gaussians* dataset and conditional normalizing flows (Figure 4) on the *two moon* datasets. On the one hand, we observe model collapse if the generative models are fully retrained on their own synthetic data (top rows). On the other hand, if the model is retrained on a mix of real and generated data (bottom rows), then the iterative process does not diverge. In Figures 3 and 4, the model is iteratively retrained on as much generated data as real data. For Figure 3 we iteratively retrain a DDPM diffusion model for 200 epochs on $10^4$ samples. For Figure 4 we iteratively retrain a conditional flow matching model for 150 epochs on $10^3$ samples.

**Experimental Setup**. For EDM we use the code, pre-trained models, and optimizers from the official implementation (specifically the `ddpmpp` architecture with unconditional sampling) and a batch size of 128 for FFHQ-64 and 256 for CIFAR-10. For OTCFM, we used the `torch-cfm` implementation. All other hyperparameters are those provided in the official implementations. After each training iteration, we generate a set of images of the same size as the training set, $\{\tilde{\mathbf{x}}_i\}_{i=1}^n$, $\tilde{\mathbf{x}}_i \sim p_{\boldsymbol{\theta}}$. This corresponds to $5 \cdot 10^4$ images for CIFAR-10 and $7 \cdot 10^4$ images for FFHQ-64. At each iteration, we compute the FID (Heusel et al., 2017) as well as precision and recall (Kynkäänniemi et al. 2019, which are correlated with fidelity and diversity) between the real data and the full set of generated data. All the metrics are computed using the FLS package (`https://github.com/marcojira/fls`, Jiralerspong et al. 2023, ee Appendix G for further details). Then, following Algorithm 1 we create a dataset which is a mix of all the real data $\mathcal{D}_{\text{real}}$, and a fraction of the generated data $\mathcal{D}_{\text{synth}} = \{\tilde{\mathbf{x}}_i\}_{i=1}^{\lfloor \lambda \cdot n \rfloor}$. *After each retraining iteration, the network and the optimizer are saved*, and *the next retraining step resumes from this checkpoint*: CIFAR-10 and FFHQ-64 models are *finetuned* for $10^6$ and $1.4 \cdot 10^6$ images, which is equivalent to 20 epochs on the real dataset. Note that since the created datasets $\mathcal{D}_{\text{real}} \cup \{\mathbf{x}_i\}_{i=1}^{\lfloor \lambda \cdot n \rfloor}$ do not have the same size, *i.e.,* their size depends on $\lambda$: $|\mathcal{D}_{\text{real}} \cup \{\mathbf{x}_i\}_{i=1}^{\lfloor \lambda \cdot n \rfloor}| = n + \lfloor \lambda \cdot n \rfloor$, we train on a *constant number of images* and not epochs which is necessary for a fair comparison between multiple fractions $\lambda$ of the incorporated generated data.

**Analysis of Results**. Iterative retraining of OTCFM and EDM are displayed in Figure 2. In particular, we report FID, precision, and recall as a function of the number of retraining iterations for various values of $\lambda$. Note that the dashed black line with circles corresponds to the baseline of retraining without synthetic data $\lambda = 0$, while the solid yellow line $\lambda = 1$ is retraining with equal amounts of synthetic and original real data. The dashed red line with triangles corresponds to retraining generative models only on their own data. Every other line is an interpolation between the settings of the black and yellow lines. As predicted by Theorem 2, if the proportion of real data is large enough, *i.e.,* the value of $\lambda$ is small enough, then the retraining procedure is stable, and models do not diverge (see for instance, the CIFAR-10 and FFHQ-64—EDM rows of Figure 2. If the fraction of used generated data $\lambda$ is too large, *e.g.,* $\lambda = 1$, we observe that FID might diverge. This provides empirical evidence

---

[4]Code can be found at `https:github.com/QB3/gen_models_dont_go_mad`.

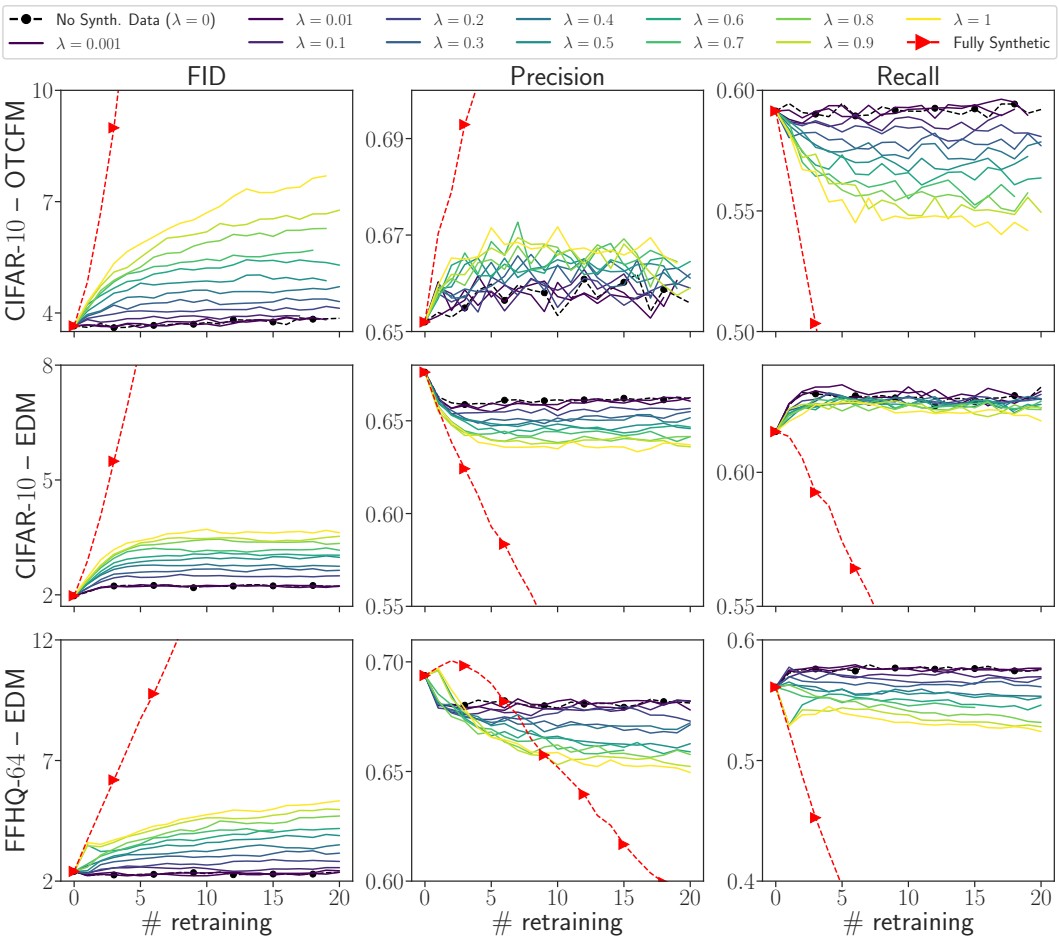

Figure 2: **FID, precision, and recall of the generative models as a function of the number of retraining for multiple fractions** $\lambda$ **of generated data,** $\mathcal{D} = \mathcal{D}_{\text{real}} \cup \{\tilde{\mathbf{x}}_i\}_{i=1}^{\lfloor \lambda \cdot n \rfloor}$, $\tilde{\mathbf{x}}_i \sim p_{\boldsymbol{\theta}_t}$. For all models and datasets, only training on synthetic data (dashed red line with triangles) yields divergence. For the EDM models on CIFAR-10 (middle row), the iterative retraining is stable for all the proportions of generated data from $\lambda = 0$ to $\lambda = 1$. For the EDM on FFHQ-64 (bottom row), the iterative retraining is stable if the proportion of used generated data is small enough ($\lambda < 0.5$).

in support of the model collapse phenomenon in both Alemohammad et al. (2023); Shumailov et al. (2023) where the generative models diverge and lead to poor sample quality. Interestingly, for all fractions $\lambda$ used to create the iterative training dataset $\mathcal{D}_t$, the precision of EDM models decreases with the number of retraining while their recall increases. It is worth noting we observe the inverse trend between precision for OTCFM on CIFAR-10.

## 6 CONCLUSION AND OPEN QUESTIONS

We investigated the iterative retraining of generative models on their own synthetic data. Our main contribution is showing that if the generative model initially trained on real data is good enough, and the iterative retraining is made on a mixture of synthetic and real data, then the retraining procedure Algorithm 1 is stable (Theorems 1 and 2). Additionally, we validate our theoretical findings (Theorems 1 and 2) empirically on natural image datasets (CIFAR-10 and FFHQ-64) with various powerful generative models (OTCFM, DDPM, and EDM). One of the limitations of Theorems 1 and 2 is that the proposed sufficient condition to ensure stability may not be necessary. Precisely, under Assumptions 1 and 2, there is a gap to close between the necessity of $\lambda < +\infty$ and the sufficiency $\lambda(1 + \frac{L\epsilon}{\alpha}) < 1$ for local stability. Another avenue for future work is to understand better the impact of the finite sampling aspect (described in Section 3.2) on iterative retaining stability.

ACKNOWLEDGEMENTS

QB would like to thank Kilian Fatras for providing guidance on the conditional flow matching experiments, Nate Gillman for fruitful discussions yielding proof clarifications, and Samsung Electronics Co., Ldt. for partially funding this research. The authors would like to thank Nvidia for providing computing resources for this research.

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

# A    PROOF OF PROPOSITION 1

$$\text{Sampling step:}\quad \left\{ \mathbf{x}_t^j = \boldsymbol{\mu}_t + \sqrt{\Sigma_t}\mathbf{z}_t^j, \text{ with } \mathbf{Z}_j^t \overset{\text{i.i.d.}}{\sim} \mathcal{N}_{0,\mathbf{I}}, \, 1 \le j \le n \right. ,  \tag{4}$$

$$\text{Learning step:}\quad \left\{ \boldsymbol{\mu}_{t+1} = \frac{1}{n}\sum_j \mathbf{x}_t^j \right. , \, \Sigma_{t+1} = \frac{1}{n-1}\sum_j \left( \mathbf{x}_t^j - \boldsymbol{\mu}_{t+1} \right)\left( \mathbf{x}_t^j - \boldsymbol{\mu}_{t+1} \right)^T.  \tag{5}$$

**Proposition 1.** *(Gaussian Collapse) For all initializations of the mean $\mu_0$ and the covariance $\Sigma_0$, iteratively learning a single multivariate Gaussian solely on its generated data yields model collapse. More precisely, if $\mu_t$ and $\Sigma_t$ follows Equations (4) and (5), then, there exists $\alpha < 1$,*

$$\mathbb{E}(\sqrt{\boldsymbol{\Sigma}_t}) \preceq \alpha^t \sqrt{\boldsymbol{\Sigma}_0} \xrightarrow[t \to +\infty]{} 0  .  \tag{6}$$

For exposition purposes, we first prove Proposition 1 for a single one-dimensional Gaussian (Appendix A.1). Then prove Proposition 1 for a single multivariate Gaussian in Appendix A.2.

## A.1    1D-GAUSSIAN CASE.

In the one-dimensional case, the sampling and the learning steps become

$$\text{Sampling step:}\quad \left\{ x_j^t = \mu_t + \sigma_t z_t^j, \text{ with } z_t^j \overset{\text{i.i.d.}}{\sim} \mathcal{N}_{0,\mathbf{I}}, \, 1 \le j \le n \right. ,  \tag{15}$$

$$\text{Learning step:}\quad \left\{ \mu_{t+1} = \frac{1}{n}\sum_j x_t^j \right. , \, \sigma_{t+1}^2 = \frac{1}{n-1}\sum_j \left( x_t^j - \mu_{t+1} \right)^2  .  \tag{16}$$

Proposition 1 can be proved using the following lemmas. First, one can obtain a recursive formula for the variances $\sigma_t^2$ (Lemma A.1 i)). Then *a key component of the proof is to take the expectation of the standard deviation $\sigma_t$ (not the variance $\sigma_t^2$, which stays constant because we used an unbiased estimator of the variance)*, this is done in Lemma A.1 ii). Finally, Lemma A.1 iii) shows how to upper-bound the expectation of the square root of a $\chi^2$ variable.

**Lemma A.1.**    i) *If $(\sigma_t)$ and $(\mu_t)$ follow Equations (15) and (16), then, for all $\mu_0, \sigma_0$,*

$$\sigma_{t+1}^2 = \frac{1}{n-1}\sum_j \left( z_t^j - \frac{1}{n}\sum_{j'} Z_{j'}^t \right)^2 \sigma_t^2  .  \tag{17}$$

ii)

$$\forall t \ge 1, \sigma_{t+1} = \frac{\sqrt{S_t}}{\sqrt{n-1}}\sigma_t  , \, \sqrt{S_t} \sim \chi_{n-1}  .  \tag{18}$$

iii)

$$\forall t \le 1, \mathbb{E}(\sigma_t) = \alpha^t \sigma_0  ,  \tag{19}$$

$$\text{with}\quad \alpha = \alpha(n) = \frac{\mathbb{E}\left( \sqrt{\chi_{n-1}^2} \right)}{\sqrt{n-1}} < 1  .$$

*Proof.* (Lemma A.1 i)). Let $t \ge 1$.

$$\mu_{t+1} = \frac{1}{n}\sum_j X_j^t = \frac{1}{n}\sum_j (\mu_t + z_t^j \sigma_t)  \tag{20}$$

$$= \mu_t + \frac{\sigma_t}{n}\sum_j z_j^t  ,  \tag{21}$$

hence

$$\sigma_{t+1}^2 = \frac{1}{n-1} \sum_j \left( X_j^t - \mu_{t+1} \right)^2 \tag{22}$$

$$= \frac{1}{n-1} \sum_j \left( \mu_t + z_t^j \sigma_t - \mu_t - \frac{\sigma_t}{n} \sum_{j'} z_{j'}^t \right)^2 \tag{23}$$

$$= \frac{1}{n-1} \sum_j \left( z_t^j - \frac{1}{n} \sum_{j'} z_{j'}^t \right)^2 \sigma_t^2 \ . \tag{24}$$

$\square$

*Proof.* (Lemma A.1 *ii)*) Cochran theorem states that $S_t := \sum_j \left( z_t^j - \frac{1}{n} \sum_{j'} z_{j'}^t \right)^2 \sim \chi_{n-1}^2$. Combined with Lemma A.1 *i)*, this yields

$$\forall t \geq 1, \sigma_{t+1} = \frac{\sqrt{S_t}}{\sqrt{n-1}} \sigma_t \ , \tag{25}$$

where $\sqrt{S_t} \sim \chi_{n-1}$.

$\square$

*Proof.* (Lemma A.1 *iii)*). The independence of the $z_t^j$ implies independence of the $S_t$, which yields that for all $t \geq 1$

$$\mathbb{E}(\sigma_{t+1}|\sigma_t) = \mathbb{E}\left( \frac{\sqrt{S_t}}{\sqrt{n-1}} | \sigma_t \right) \sigma_t \tag{26}$$

$$= \mathbb{E}\left( \frac{\sqrt{S_t}}{\sqrt{n-1}} \right) \sigma_t \ \text{ because } S_t \perp\!\!\!\perp \sigma_t \ , \tag{27}$$

$$\mathbb{E}(\sigma_{t+1}) = \underbrace{\mathbb{E}\left( \frac{\sqrt{S_t}}{\sqrt{n-1}} \right)}_{:=\alpha} \mathbb{E}(\sigma_t) \ \text{ because } \mathbb{E}(\mathbb{E}(\sigma_{t+1}|\sigma_t)) = \mathbb{E}(\sigma_{t+1}) \ . \tag{28}$$

The independence of the $Z^t$ yields independence of the $S_t$. Combined with the strict Jensen inequality applied to the square root, it yields

$$\alpha = \mathbb{E}\left( \frac{\sqrt{S_t}}{\sqrt{n-1}} \right) \tag{29}$$

$$< \sqrt{\mathbb{E}\left( \frac{S_t}{n-1} \right)} \tag{30}$$

$$= 1 \ \text{ because } \mathbb{E}(S_t) = n-1 \text{ since } S_t \sim \chi_{n-1}^2 \ . \tag{31}$$

The strict Jensen inequality comes from the fact that the square root function is strictly convex, and $S_t \sim \chi_{n-1}^2$ is not a constant random variable. Note that $\alpha$ is strictly smaller than 1 and is independent of the number of retraining. However $\alpha$ depends on the number of samples $n$, and that $\alpha(n) \to 1$, when $n \to \infty$.

$\square$

## A.2 MULTI-DIMENSIONAL CASE.

Sampling step: $\quad \left\{ \mathbf{x}_t^j = \boldsymbol{\mu}_t + \sqrt{\boldsymbol{\Sigma}_t} \mathbf{z}_t^j, \text{ with } \mathbf{z}_t^j \overset{\text{i.i.d.}}{\sim} \mathcal{N}(0, \mathbf{I}), 1 \leq j \leq n \ , \right.$ $\tag{4}$

Learning step: $\quad \left\{ \boldsymbol{\mu}_{t+1} = \frac{1}{n} \sum_j \mathbf{x}_t^j \ , \ \boldsymbol{\Sigma}_{t+1} = \frac{1}{n-1} \sum_{\mathbf{j}} \left( \mathbf{x}_{\mathbf{t}}^{\mathbf{j}} - \boldsymbol{\mu}_{\mathbf{t+1}} \right) \left( \mathbf{x}_{\mathbf{t}}^{\mathbf{j}} - \boldsymbol{\mu}_{\mathbf{t+1}} \right)^{\mathbf{T}} . \right.$ $\tag{5}$

The generalization to the multi-dimensional follows the same scheme as the unidimensional case: the key point is to upper the expectation of the *square root of the covariance matrix*

**Lemma A.2.**    i) *If* $(\mathbf{\Sigma}_t)$ *and* $(\boldsymbol{\mu}_t)$ *follow Equations (4) and (5), then, for all* $\boldsymbol{\mu}_0, \mathbf{\Sigma}_0$,

$$\mathbf{\Sigma}_{t+1} = \frac{1}{n-1} \sqrt{\mathbf{\Sigma}_t} \underbrace{\left[ \sum_j \left( \mathbf{z}_t^j - \frac{1}{n} \sum_{j'} \mathbf{Z}_{j'}^t \right) \left( \mathbf{z}_t^j - \frac{1}{n} \sum_{j'} \mathbf{Z}_{j'}^t \right)^\top \right]}_{:=\mathbf{S}} \sqrt{\mathbf{\Sigma}_t} \ . \quad (32)$$

ii) *Let*

$$\mathbf{S} = \sum_j \left( \mathbf{z}_t^j - \frac{1}{n} \sum_{j'} \mathbf{Z}_{j'}^t \right) \left( \mathbf{z}_t^j - \frac{1}{n} \sum_{j'} \mathbf{Z}_{j'}^t \right)^\top \ , \quad (33)$$

*then*

$$\sqrt{\mathbb{E}(\mathbf{S})} = \sqrt{n-1}\mathbf{I}_d \ , \quad (34)$$

iii) *Then*

$$\mathbb{E}(\sqrt{\mathbf{S}}) \preceq \alpha \sqrt{n-1}\mathbf{I}_d = \sqrt{\mathbb{E}(\mathbf{S})} \ , \quad (35)$$

*with* $0 \le \alpha = \lambda_{\max}\left( \frac{\mathbb{E}(\sqrt{\mathbf{S}})}{\sqrt{n-1}} \right) < 1.$

iv) *Finally*

$$\mathbb{E}\left( \sqrt{\mathbf{\Sigma}_{t+1}} \right) \preceq \alpha \mathbb{E}\left( \sqrt{\mathbf{\Sigma}_t} \right) \ , \quad (36)$$

*with* $0 \le \alpha < 1.$

*Proof.* (Lemma A.2 *i)*) Rearranging the learning and the sampling steps (Equations (4) and (5)) yields

$$\mathbf{x}_t^j = \boldsymbol{\mu}_t + \sqrt{\mathbf{\Sigma}_t}\mathbf{z}_t^j \ , \quad (37)$$

$$\boldsymbol{\mu}_{t+1} = \frac{1}{n} \sum_j \mathbf{x}_t^j \ , \quad (38)$$

$$\boldsymbol{\mu}_{t+1} = \boldsymbol{\mu}_t + \frac{1}{n} \sqrt{\mathbf{\Sigma}_t} \sum_j \mathbf{z}_t^j \ , \text{ hence} \quad (39)$$

$$\mathbf{x}_t^j - \boldsymbol{\mu}_{t+1} = \sqrt{\mathbf{\Sigma}_t} \left( \mathbf{z}_t^j - \frac{1}{n} \sum_j \mathbf{z}_t^j \right) \ . \quad (40)$$

Plugging Equation (40) in the learning step (Equation (5)) yields

$$\mathbf{\Sigma}_{t+1} = \frac{1}{n-1} \sqrt{\mathbf{\Sigma}_t} \left[ \sum_j \left( \mathbf{z}_t^j - \frac{1}{n} \sum_{j'} \mathbf{Z}_{j'}^t \right) \left( \mathbf{z}_t^j - \frac{1}{n} \sum_{j'} \mathbf{Z}_{j'}^t \right)^\top \right] \sqrt{\mathbf{\Sigma}_t} \ . \quad (41)$$

$\square$

*Proof.* ([Lemma A.2 *ii)*) Since $\mathbf{Z}_j^t \overset{\text{iid}}{\sim} \mathcal{N}(0, \mathbf{I})$, then

$$\mathbb{E}(\mathbf{S}) = \mathbb{E}\left( \sum_j \left( \mathbf{z}_t^j - \frac{1}{n}\sum_{j'} \mathbf{Z}_{j'}^t \right) \left( \mathbf{z}_t^j - \frac{1}{n}\sum_{j'} \mathbf{Z}_{j'}^t \right)^\top \right) \ , \tag{42}$$

$$= \sum_j \mathbb{E}\left( \mathbf{z}_t^j - \frac{1}{n}\sum_{j'} \mathbf{Z}_{j'}^t \right) \left( \mathbf{z}_t^j - \frac{1}{n}\sum_{j'} \mathbf{Z}_{j'}^t \right)^\top \ , \tag{43}$$

$$= \sum_j \mathbb{E}\left( \mathbf{z}_t^j - \frac{1}{n}\sum_{j'} \mathbf{Z}_{j'}^t \right)^2 \cdot \mathbf{I} \ , \tag{44}$$

$$= \underbrace{\mathbb{E}\sum_j \left( \mathbf{z}_t^j - \frac{1}{n}\sum_{j'} \mathbf{Z}_{j'}^t \right)^2}_{\text{expectation of a unidimensional } \chi^2_{n-1} \text{ variable}} \cdot \mathbf{I} \ , \tag{45}$$

$$= (n-1) \cdot \mathbf{I} \ . \tag{46}$$

$\square$

*Proof.* ([Lemma A.2 *iii)*) The matrix square root operator is strictly concave ([Marshall et al., 1979](), Chapter 16, Example E.7.d), in addition, $\mathbf{S}$ is not a Dirac, hence the strict Jensen inequality yields

$$\mathbb{E}(\sqrt{\mathbf{S}}) \prec \sqrt{\mathbb{E}(\mathbf{S})} \ , \tag{47}$$

$$\prec \sqrt{n-1} \cdot \mathbf{I}_d \text{ (by Lemma A.2 *ii)*) } \ , \tag{48}$$

$$\preceq \alpha\sqrt{n-1} \cdot \mathbf{I}_d \ , \tag{49}$$

with $= \alpha = \lambda_{\max}\left( \frac{\mathbb{E}(\sqrt{\mathbf{S}})}{\sqrt{n-1}} \right) < 1.$ $\square$

*Proof.* ([Lemma A.2 *iv)*)

Taking the matrix square root of [Equation (41)]() yields

$$\sqrt{\mathbf{\Sigma}_{t+1}} = \frac{1}{n-1}\mathbf{\Sigma}_t^{\frac{1}{4}} \underbrace{\sqrt{\left[ \sum_j \left( \mathbf{z}_t^j - \frac{1}{n}\sum_{j'} \mathbf{Z}_{j'}^t \right) \left( \mathbf{z}_t^j - \frac{1}{n}\sum_{j'} \mathbf{Z}_{j'}^t \right)^\top \right]}}_{=\sqrt{\mathbf{S}}} \mathbf{\Sigma}_t^{\frac{1}{4}} \ , \tag{50}$$

$$\sqrt{\mathbf{\Sigma}_{t+1}} = \frac{1}{n-1}\mathbf{\Sigma}_t^{\frac{1}{4}} \sqrt{\mathbf{S}} \mathbf{\Sigma}_t^{\frac{1}{4}} \ . \tag{51}$$

Taking the expectation of [Equation (51)]() conditioned on $\mathbf{\Sigma}_t$ yields

$$\mathbb{E}\left( \sqrt{\mathbf{\Sigma}_{t+1}} | \mathbf{\Sigma}_t \right) = \frac{1}{\sqrt{n-1}}\mathbf{\Sigma}_t^{\frac{1}{4}} \mathbb{E}\left( \sqrt{\mathbf{S}} | \mathbf{\Sigma}_t \right) \mathbf{\Sigma}_t^{\frac{1}{4}} \ , \tag{52}$$

$$= \frac{1}{\sqrt{n-1}}\mathbf{\Sigma}_t^{\frac{1}{4}} \mathbb{E}\left( \sqrt{\mathbf{S}} \right) \mathbf{\Sigma}_t^{\frac{1}{4}} \text{ (because } \mathbf{S} \perp\!\!\!\perp \mathbf{\Sigma}_t) \ , \tag{53}$$

$$\preceq \frac{1}{\sqrt{n-1}}\mathbf{\Sigma}_t^{\frac{1}{4}} \alpha\sqrt{\mathbb{E}(\mathbf{S})} \mathbf{\Sigma}_t^{\frac{1}{4}} \text{ (by Lemma A.2 *iii)*) } \ , \tag{54}$$

$$= \alpha\frac{1}{\sqrt{n-1}}\mathbf{\Sigma}_t^{\frac{1}{4}} \sqrt{(n-1)\mathbf{I}}\mathbf{\Sigma}_t^{\frac{1}{4}} \ , \tag{55}$$

$$= \alpha\sqrt{\mathbf{\Sigma}_{+1}} \ , \tag{56}$$

$$\mathbb{E}\left( \sqrt{\mathbf{\Sigma}_{t+1}} \right) \preceq \alpha\mathbb{E}\left( \sqrt{\mathbf{\Sigma}_t} \right) \text{ (because } \mathbb{E}\left( \mathbb{E}\left( \sqrt{\mathbf{\Sigma}_{t+1}} | \mathbf{\Sigma}_t \right) \right) = \mathbb{E}\left( \sqrt{\mathbf{\Sigma}_{t+1}} \right)) \ . \tag{57}$$

$\square$

## B PROOF OF PROPOSITION 2

**Proposition 2.** *Let $p_{\boldsymbol{\theta}} : \mathbf{x} \mapsto \mathcal{N}_{\boldsymbol{\mu}, \boldsymbol{\Sigma}}(\mathbf{x}) = e^{-\frac{1}{2}(\mathbf{x}-\boldsymbol{\mu})^{\top} \boldsymbol{\Sigma}^{-1}(\mathbf{x}-\boldsymbol{\mu})} / \sqrt{(2\pi)^d |\boldsymbol{\Sigma}|}$, with $\boldsymbol{\mu} \in \mathbb{R}^d$, $\boldsymbol{\Sigma} \succ 0$, $\boldsymbol{\theta} = (\boldsymbol{\mu}, \boldsymbol{\Sigma}^{-1})$, then $\mathbb{E}_{\mathbf{x} \sim p_{\boldsymbol{\theta}}(\mathbf{x})}[\nabla_{\boldsymbol{\theta}}^2 \log p_{\boldsymbol{\theta}}(\mathbf{x})] \prec 0$ and $\mathbf{x} \mapsto \nabla_{\boldsymbol{\theta}}^2 \log p_{\boldsymbol{\theta}}(\mathbf{x})$ is 1-Lipschitz.*

*(Proof of Proposition 2.)* Following the computation of Barfoot (2020), and using the fact that $\nabla_{\boldsymbol{\Sigma}^{-1}} \log \det(\boldsymbol{\Sigma}) = \boldsymbol{\Sigma}$ (Boyd and Vandenberghe, 2004, Sec A.4.1) one has

$$-\log p_{\boldsymbol{\theta}}(\mathbf{x}) = -\frac{1}{2} \log \det(\boldsymbol{\Sigma}^{-1}) + \frac{1}{2}(\mathbf{x} - \boldsymbol{\mu})^{\top} \boldsymbol{\Sigma}^{-1}(\mathbf{x} - \boldsymbol{\mu}) + \frac{d}{2} \log(2\pi) \text{ , hence} \quad (58)$$

$$-\nabla_{(\boldsymbol{\mu}, \boldsymbol{\Sigma})} \log p_{\boldsymbol{\theta}}(\mathbf{x}) = \begin{pmatrix} \boldsymbol{\Sigma}^{-1}(\boldsymbol{\mu} - \mathbf{x}) \\ \frac{1}{2}(\mathbf{x} - \boldsymbol{\mu})(\mathbf{x} - \boldsymbol{\mu})^{\top} - \boldsymbol{\Sigma} \end{pmatrix} \text{ , then} \quad (59)$$

$$-\nabla_{(\boldsymbol{\mu}, \boldsymbol{\Sigma})}^2 \log p_{\boldsymbol{\theta}}(\mathbf{x}) = \begin{pmatrix} \boldsymbol{\Sigma}^{-1} & (\boldsymbol{\mu} - \mathbf{x}) \otimes \mathbf{I}_d \\ \mathbf{I}_d \otimes (\boldsymbol{\mu} - \mathbf{x}) & \frac{1}{2}(\boldsymbol{\Sigma} \otimes \boldsymbol{\Sigma}) \end{pmatrix} \text{ .} \quad (60)$$

Equation (60) shows that $\mathbf{x} \mapsto \nabla_{\boldsymbol{\theta}}^2 \log p_{\boldsymbol{\theta}}(\mathbf{x})$ is 1-Lipschitz. In addition, since $\boldsymbol{\Sigma} \succ 0$, then $\mathbb{E}[-\nabla_{(\boldsymbol{\mu}, \boldsymbol{\Sigma}^{-1})}^2 \log p_{\boldsymbol{\theta}}(\mathbf{x})] \succ 0$. $\square$

## C PROOF OF PROPOSITION 3

**Proposition 3.** *(The Local Maximum Likelihood Solution is Unique) Let $\boldsymbol{\theta}^{\star}$ be a solution of Equation (7) that satisfies Assumptions 1 and 2. If $\epsilon L < \alpha$, then for all $\lambda > 0$ and $\boldsymbol{\theta}$ in a small enough neighborhood $\mathcal{U}$ around $\boldsymbol{\theta}^{\star}$, there exists a unique local maximizer $\mathcal{G}_{\lambda}^{\infty}(\boldsymbol{\theta})$ in $\mathcal{U}$.*

*(Proof of Proposition 3).* Let $\boldsymbol{\theta}^{\star}$ be a solution of Equation (7). Assume that $\boldsymbol{\theta}^{\star}$ follows Assumptions 1 and 2. We recall that

$$\mathcal{H}(\boldsymbol{\theta}, \boldsymbol{\theta}') := \overbrace{\mathbb{E}_{\mathbf{x} \sim p_{\text{data}}}[\log p_{\boldsymbol{\theta}'}(\mathbf{x})]}^{:= \mathcal{H}_1(\boldsymbol{\theta}')} + \lambda \overbrace{\mathbb{E}_{\tilde{\mathbf{x}} \sim p_{\boldsymbol{\theta}}}[\log p_{\boldsymbol{\theta}'}(\tilde{\mathbf{x}})]}^{:= \mathcal{H}_2(\boldsymbol{\theta}, \boldsymbol{\theta}')} \text{ , and} \quad (61)$$

$$\mathcal{G}_{\lambda}^{\infty}(\boldsymbol{\theta}) := \underset{\boldsymbol{\theta}'}{\operatorname{argmax}} \, \mathcal{H}(\boldsymbol{\theta}, \boldsymbol{\theta}') \text{ .} \quad (62)$$

Using Proposition 4, since $\boldsymbol{\theta}^{\star}$ is a solution of Equation (7), then $\nabla_{\boldsymbol{\theta}'} \mathcal{H}(\boldsymbol{\theta}^{\star}, \boldsymbol{\theta}^{\star}) = 0$. In addition, Assumption 2 ensures that $\boldsymbol{\theta}^{\star} \in \text{local-argmax}_{\boldsymbol{\theta}' \in \Theta} \mathcal{H}(\boldsymbol{\theta}^{\star}, \boldsymbol{\theta}')$ and that $\nabla_{\boldsymbol{\theta}'}^2 \mathcal{H}(\boldsymbol{\theta}^{\star}, \boldsymbol{\theta}^{\star})$ is invertible. Hence, by the implicit function theorem (Lang, 1999, Theorem 5.9), for $\boldsymbol{\theta}$ in a neighborhood of $\boldsymbol{\theta}^{\star}$ we have that there exists a continuous function $g$ such that $g(\boldsymbol{\theta}^{\star}) = \boldsymbol{\theta}^{\star}$ and $\nabla_{\boldsymbol{\theta}'} \mathcal{H}(\boldsymbol{\theta}, g(\boldsymbol{\theta})) = 0$. Finally, in order to show that $g(\boldsymbol{\theta})$ is a local maximizer of $\boldsymbol{\theta}' \mapsto \mathcal{H}(\boldsymbol{\theta}, \boldsymbol{\theta}')$ we just need to show that $\nabla_{\boldsymbol{\theta}'}^2 \mathcal{H}(\boldsymbol{\theta}, \boldsymbol{\theta}^{\star}) \prec 0$.

$$\nabla_{\boldsymbol{\theta}'}^2 \mathcal{H}(\boldsymbol{\theta}, \boldsymbol{\theta}^{\star}) = \nabla_{\boldsymbol{\theta}'}^2 \mathcal{H}_1(\boldsymbol{\theta}^{\star}) + \lambda \nabla_{\boldsymbol{\theta}'}^2 \mathcal{H}_2(\boldsymbol{\theta}, \boldsymbol{\theta}^{\star}) \quad (63)$$

$$= (1 + \lambda) \underbrace{\nabla_{\boldsymbol{\theta}'}^2 \mathcal{H}_1(\boldsymbol{\theta}^{\star})}_{\preceq -\alpha \mathbf{I} \text{ (using Assumption 2)}} + \lambda (\nabla_{\boldsymbol{\theta}'}^2 \mathcal{H}_2(\boldsymbol{\theta}, \boldsymbol{\theta}^{\star}) - \nabla_{\boldsymbol{\theta}'}^2 \mathcal{H}_1(\boldsymbol{\theta}^{\star})) \text{ ,} \quad (64)$$

$$\preceq -(1 + \lambda)\alpha \mathbf{I} + \lambda \underbrace{(\mathbb{E}_{\mathbf{x} \sim p_{\boldsymbol{\theta}}}[\nabla^2 \log p_{\boldsymbol{\theta}'}(\mathbf{x})] - \mathbb{E}_{\mathbf{x} \sim p_{\text{data}}}[\nabla^2 \log p_{\boldsymbol{\theta}'}(\mathbf{x})])}_{\preceq \varepsilon L \text{ (using Assumption 1)}} \text{ ,} \quad (65)$$

$$\preceq -\alpha(1 + \lambda)\mathbf{I}_d + \lambda \varepsilon L \mathbf{I}_d \text{ ,} \quad (66)$$

$$\prec 0 \text{ if } \alpha \geq \varepsilon L \text{ .} \quad (67)$$

$\square$

## D PROOF OF PROPOSITION 4

**Proposition 4.** *(The Optimal Parametric Generative Model is a Fixed Point) For a given data distribution $p_{\text{data}}$, any $\boldsymbol{\theta}^{\star}$ solution of Equation (7), and for all $\lambda > 0$ we have,*

$$\boldsymbol{\theta}^{\star} = \mathcal{G}_{\lambda}^{\infty}(\boldsymbol{\theta}^{\star}) \text{ .} \quad (10)$$

*Proof.* We recall the definition of $\boldsymbol{\theta}^\star$ and $\mathcal{G}_\lambda^\infty$ (Equations (7) and (9)):

$$\boldsymbol{\theta}^\star \in \underset{\boldsymbol{\theta}' \in \boldsymbol{\theta}}{\operatorname{argmax}} \, \mathbb{E}_{\mathbf{x} \sim p_{\text{data}}}[\log p_{\boldsymbol{\theta}'}(\mathbf{x})] \ . \tag{7}$$

$$\mathcal{G}_\lambda^\infty(\boldsymbol{\theta}) \in \underset{\boldsymbol{\theta}' \in \boldsymbol{\theta}}{\operatorname{local\text{-}argmax}}(\underbrace{\mathbb{E}_{\mathbf{x} \sim p_{\text{data}}}[\log p_{\boldsymbol{\theta}'}(\mathbf{x})]}_{:= \mathcal{H}_1(\boldsymbol{\theta}')} + \lambda \underbrace{\mathbb{E}_{\tilde{\mathbf{x}} \sim p_{\boldsymbol{\theta}}}[\log p_{\boldsymbol{\theta}'}(\tilde{\mathbf{x}})]}_{:= \mathcal{H}_2(\boldsymbol{\theta}, \boldsymbol{\theta}')}) \ . \tag{9}$$

Since $\boldsymbol{\theta}^\star$ is a solution of Equation (7)

$$\forall \boldsymbol{\theta}' \in \Theta, \mathcal{H}_1(\boldsymbol{\theta}') = \mathbb{E}_{\mathbf{x} \sim p_{\text{data}}}[\log p_{\boldsymbol{\theta}'}(\mathbf{x})] \tag{68}$$

$$\leq \mathbb{E}_{\mathbf{x} \sim p_{\text{data}}}[\log p_{\boldsymbol{\theta}^\star}(\mathbf{x})] \tag{69}$$

$$\leq \mathcal{H}_1(\boldsymbol{\theta}^\star) \ . \tag{70}$$

Gibbs inequality yields

$$\forall \boldsymbol{\theta}' \in \Theta, \ \ \mathcal{H}_2(\boldsymbol{\theta}^\star, \boldsymbol{\theta}') = \mathbb{E}_{\tilde{\mathbf{x}} \sim p_{\boldsymbol{\theta}^\star}}[\log p_{\boldsymbol{\theta}'}(\tilde{\mathbf{x}})] \tag{71}$$

$$\leq \mathbb{E}_{\tilde{\mathbf{x}} \sim p_{\boldsymbol{\theta}^\star}}[\log p_{\boldsymbol{\theta}^\star}(\tilde{\mathbf{x}})] \tag{72}$$

$$\leq \mathcal{H}_2(\boldsymbol{\theta}^\star, \boldsymbol{\theta}^\star) \ . \tag{73}$$

$\mathcal{H}_1$ and $\mathcal{H}_2(\boldsymbol{\theta}^\star, \cdot)$ are both maximized in $\boldsymbol{\theta}^\star$ hence

$$\mathcal{G}_\lambda^\infty(\boldsymbol{\theta}^\star) = \underset{\boldsymbol{\theta}'}{\operatorname{argmax}} \, \mathcal{H}_1(\boldsymbol{\theta}') + \lambda \mathcal{H}_2(\boldsymbol{\theta}^\star, \boldsymbol{\theta}') = \boldsymbol{\theta}^\star \ . \tag{74}$$

$\square$

# E   PROOF OF THEOREM 1

$$\boldsymbol{\theta}^\star \in \operatorname*{argmax}_{\boldsymbol{\theta}' \in \boldsymbol{\theta}} \mathbb{E}_{\mathbf{x} \sim p_{\text{data}}}[\log p_{\boldsymbol{\theta}'}(\mathbf{x})] \ . \tag{7}$$

$$\mathcal{G}_\lambda^\infty(\boldsymbol{\theta}) \in \operatorname*{local-argmax}_{\boldsymbol{\theta}' \in \boldsymbol{\theta}}(\underbrace{\mathbb{E}_{\mathbf{x} \sim p_{\text{data}}}[\log p_{\boldsymbol{\theta}'}(\mathbf{x})]}_{:=\mathcal{H}_1(\boldsymbol{\theta}')} + \lambda \underbrace{\mathbb{E}_{\tilde{\mathbf{x}} \sim p_{\boldsymbol{\theta}}}[\log p_{\boldsymbol{\theta}'}(\tilde{\mathbf{x}})]}_{:=\mathcal{H}_2(\boldsymbol{\theta}, \boldsymbol{\theta}')}) \ . \tag{9}$$

**Assumption 1.** *For $\boldsymbol{\theta}$ close enough to $\boldsymbol{\theta}^\star$, the mapping $x \mapsto \nabla_{\boldsymbol{\theta}}^2 \log p_{\boldsymbol{\theta}}(\mathbf{x})$ is L-Lipschitz.*

**Assumption 2.** *The mapping $\boldsymbol{\theta} \mapsto \mathbb{E}_{\mathbf{x} \sim p_{\text{data}}}[\log p_{\boldsymbol{\theta}}(\mathbf{x})]$ is continuously twice differentiable locally around $\boldsymbol{\theta}^\star$ and $\mathbb{E}_{\mathbf{x} \sim p_{\text{data}}}[\nabla_{\boldsymbol{\theta}}^2 \log p_{\boldsymbol{\theta}^\star}(\mathbf{x})] \preceq -\alpha \mathbf{I}_d \prec 0$.*

**Theorem 1.** *(Stability of Iterative Retraining) Given $\boldsymbol{\theta}^\star$ as defined in Equation (7) that follows Assumptions 1 and 2, we have that, if $\lambda(1 + \frac{L\varepsilon}{\alpha}) < 1/2$, then the operator norm of the Jacobian is strictly bounded by 1, more precisely,*

$$\|\mathcal{J}_{\boldsymbol{\theta}} \mathcal{G}_\lambda^\infty(\boldsymbol{\theta}^\star)\|_2 \leq \frac{\lambda(\alpha + \varepsilon L)}{\alpha - \lambda(\alpha + \varepsilon L)} < 1 \ . \tag{11}$$

*Consequently, there exists a $R > 0$ such that for all $\boldsymbol{\theta}_0 \in \boldsymbol{\theta}$ that satisfy $\|\boldsymbol{\theta}_0 - \boldsymbol{\theta}^\star\| \leq R$, then starting at $\boldsymbol{\theta}_0$ and having $\boldsymbol{\theta}_{t+1} = \mathcal{G}_\lambda^\infty(\boldsymbol{\theta}_t)$ we have that $\boldsymbol{\theta}_t \xrightarrow[t \to +\infty]{} \boldsymbol{\theta}^\star$ and*

$$\|\boldsymbol{\theta}_t - \boldsymbol{\theta}^\star\| = \tilde{O}\left(\left(\frac{\lambda(\alpha + \varepsilon L)}{\alpha - \lambda(\alpha + \varepsilon L)}\right)^t \|\boldsymbol{\theta}_0 - \boldsymbol{\theta}^\star\|\right) \ . \tag{12}$$

The main goal of Theorem 1 is to show that the operator norm of the Jacobian of the fixed-point operator $\mathcal{G}$ is strictly bounded by one. We prove Theorem 1 with the following steps: using the Implicit Function Theorem, Schwarz theorem, and analytic manipulations (Lemmas E.1 *i)* and E.1 *ii)*) we managed to obtain a simple formula for the Jacobian of $\mathcal{G}$ at $\boldsymbol{\theta}^\star$. Using Kantorovich-Rubenstein duality, we manage to bound the Jacobian of the fixed-point operator at $\boldsymbol{\theta}^\star$ (Lemma E.1 *iii)*), and thus to provide a condition for which $\|\mathcal{J}\mathcal{G}_\lambda(\boldsymbol{\theta}^\star)\|_2 < 1$. These steps are detailed formally in Lemma E.1.

**Lemma E.1.** *With $\mathbf{A} = \nabla_{\boldsymbol{\theta}', \boldsymbol{\theta}'}^2 \mathcal{H}_1(\boldsymbol{\theta}^\star)$ and $\mathbf{B} = \nabla_{\boldsymbol{\theta}, \boldsymbol{\theta}'}^2 \mathcal{H}_2(\boldsymbol{\theta}^\star, \boldsymbol{\theta}^\star)$, under the assumptions of Theorem 1:*

i) *There exists an open set $\mathcal{U} \subset \Theta$ containing $\boldsymbol{\theta}^\star$ such that $\forall \boldsymbol{\theta} \in \mathcal{U}$,*

$$\mathcal{J}\mathcal{G}(\boldsymbol{\theta}) = -\lambda \left(\nabla_{\boldsymbol{\theta}', \boldsymbol{\theta}'}^2 \mathcal{H}_1(\boldsymbol{\theta}) + \lambda \nabla_{\boldsymbol{\theta}', \boldsymbol{\theta}'}^2 \mathcal{H}_2(\boldsymbol{\theta}, \mathcal{G}(\boldsymbol{\theta}))\right)^{-1} \cdot \nabla_{\boldsymbol{\theta}, \boldsymbol{\theta}'}^2 \mathcal{H}_2(\boldsymbol{\theta}, \mathcal{G}(\boldsymbol{\theta})) \ . \tag{75}$$

ii)

$$\nabla_{\boldsymbol{\theta}'\boldsymbol{\theta}'} \mathcal{H}_2(\boldsymbol{\theta}^\star, \boldsymbol{\theta}^\star) = -\nabla_{\boldsymbol{\theta}'\boldsymbol{\theta}} \mathcal{H}_2(\boldsymbol{\theta}^\star, \boldsymbol{\theta}^\star) \ , \tag{76}$$

*and thus Jacobian at $\boldsymbol{\theta}^\star$ can be written*

$$\mathcal{J}\mathcal{G}(\boldsymbol{\theta}^\star) = \left(\mathbf{I}_d + \lambda \mathbf{A}^{-1} \mathbf{B}\right)^{-1} \cdot \lambda \mathbf{A}^{-1} \mathbf{B} \ . \tag{77}$$

iii) *The spectral norm of $\mathbf{A}^{-1}\mathbf{B}$ and $\mathbf{B} - \mathbf{A}$ can be bounded*

$$\|\mathbf{A}^{-1}\mathbf{B}\|_2 \leq 1 + \frac{L\varepsilon}{\alpha} \ , \tag{78}$$

*and using Kantorovich-Rubenstein duality theorem*

$$\|\mathbf{B} - \mathbf{A}\|_2 \leq L d_W(p_{\boldsymbol{\theta}^\star}, p_{\text{data}}) \ . \tag{79}$$

*Proof.* (Lemma E.1 *i)*) The definition of $\mathcal{G}$ yields

$$\nabla_{\boldsymbol{\theta}'} \mathcal{H}(\boldsymbol{\theta}, \mathcal{G}(\boldsymbol{\theta})) = 0 \ . \tag{80}$$

Differentiating Equation (80) using the chain rule yields

$$\nabla^2_{\boldsymbol{\theta}',\boldsymbol{\theta}} \mathcal{H}(\boldsymbol{\theta}, \mathcal{G}(\boldsymbol{\theta})) = -\nabla^2_{\boldsymbol{\theta}',\boldsymbol{\theta}'} \mathcal{H}(\boldsymbol{\theta}, \mathcal{G}(\boldsymbol{\theta})) \cdot \mathcal{JG}(\boldsymbol{\theta}) \ , \tag{81}$$

which implies

$$\mathcal{JG}(\boldsymbol{\theta}) = - \left( \nabla^2_{\boldsymbol{\theta}',\boldsymbol{\theta}'} \mathcal{H}(\boldsymbol{\theta}, \mathcal{G}(\boldsymbol{\theta})) \right)^{-1} \nabla^2_{\boldsymbol{\theta}',\boldsymbol{\theta}} \mathcal{H}(\boldsymbol{\theta}, \mathcal{G}(\boldsymbol{\theta})) \ . \tag{82}$$

Since $\mathcal{H}_1$ does not depend on $\boldsymbol{\theta}$

$$\nabla_{\boldsymbol{\theta}} \mathcal{H}_1(\mathcal{G}(\boldsymbol{\theta})) = 0 \ . \tag{83}$$

Combining Equations (82) and (83) and the fact that $\mathcal{G}(\boldsymbol{\theta}^\star) = \boldsymbol{\theta}^\star$ yields

$$\mathcal{JG}(\boldsymbol{\theta}^\star) = -\lambda \left( \nabla_{\boldsymbol{\theta}',\boldsymbol{\theta}'} \mathcal{H}_1(\boldsymbol{\theta}^\star) + \lambda \nabla_{\boldsymbol{\theta}',\boldsymbol{\theta}'} \mathcal{H}_2(\boldsymbol{\theta}^\star, \boldsymbol{\theta}^\star) \right)^{-1} \nabla_{\boldsymbol{\theta}',\boldsymbol{\theta}} \mathcal{H}_2(\boldsymbol{\theta}^\star, \boldsymbol{\theta}^\star) \ . \tag{84}$$

$\square$

*Proof.* (Lemma E.1 *ii)*) First, let's compute $\nabla^2_{\boldsymbol{\theta}',\boldsymbol{\theta}} \mathcal{H}_2(\boldsymbol{\theta}, \boldsymbol{\theta}')$:

$$\mathcal{H}_2(\boldsymbol{\theta}, \boldsymbol{\theta}') = \int_X \log p_{\boldsymbol{\theta}'}(\mathbf{x}) p_{\boldsymbol{\theta}}(\mathbf{x}) \mathrm{d}\mathbf{x} \ , \text{ which yields} \tag{85}$$

$$\nabla_{\boldsymbol{\theta}'} \mathcal{H}_2(\boldsymbol{\theta}, \boldsymbol{\theta}') = \int_X \nabla_{\boldsymbol{\theta}'} \log p_{\boldsymbol{\theta}'}(\mathbf{x}) p_{\boldsymbol{\theta}}(\mathbf{x}) \mathrm{d}\mathbf{x} \ , \text{ and thus} \tag{86}$$

$$\nabla^2_{\boldsymbol{\theta}',\boldsymbol{\theta}} \mathcal{H}_2(\boldsymbol{\theta}, \boldsymbol{\theta}') = \int_X \nabla_{\boldsymbol{\theta}'} \log p_{\boldsymbol{\theta}'}(\mathbf{x}) \nabla_{\boldsymbol{\theta}} p_{\boldsymbol{\theta}}(\mathbf{x}) \mathrm{d}\mathbf{x} \ , \text{ (using Schwarz theorem).} \tag{87}$$

This yields

$$\nabla^2_{\boldsymbol{\theta}',\boldsymbol{\theta}} \mathcal{H}_2(\boldsymbol{\theta}^\star, \boldsymbol{\theta}^\star) = \int_X \nabla_{\boldsymbol{\theta}} \log p_{\boldsymbol{\theta}^\star}(\mathbf{x}) \nabla_{\boldsymbol{\theta}} p_{\boldsymbol{\theta}^\star}(\mathbf{x}) \mathrm{d}\mathbf{x} \tag{88}$$

$$= \int_X \nabla_{\boldsymbol{\theta}} [p_{\boldsymbol{\theta}^\star}(\mathbf{x}) \underbrace{\nabla_{\boldsymbol{\theta}} \log p_{\boldsymbol{\theta}^\star}(\mathbf{x})}_{= \frac{\nabla_{\boldsymbol{\theta}} p_{\boldsymbol{\theta}^\star}(\mathbf{x})}{p_{\boldsymbol{\theta}^\star}(\mathbf{x})}}] \mathrm{d}\mathbf{x} - \underbrace{\int_X \nabla^2_{\boldsymbol{\theta},\boldsymbol{\theta}} \log p_{\boldsymbol{\theta}^\star}(\mathbf{x}) p_{\boldsymbol{\theta}^\star}(\mathbf{x}) \mathrm{d}\mathbf{x}}_{= \nabla^2_{\boldsymbol{\theta}',\boldsymbol{\theta}'} \mathcal{H}_2(\boldsymbol{\theta}^\star, \boldsymbol{\theta}^\star)} \tag{89}$$

$$= \int_X \nabla^2_{\boldsymbol{\theta},\boldsymbol{\theta}} p_{\boldsymbol{\theta}^\star}(\mathbf{x}) \mathrm{d}\mathbf{x} - \nabla^2_{\boldsymbol{\theta}',\boldsymbol{\theta}'} \mathcal{H}_2(\boldsymbol{\theta}^\star, \boldsymbol{\theta}^\star) \tag{90}$$

$$= \underbrace{\nabla^2_{\boldsymbol{\theta}} \underbrace{\int_X p_{\boldsymbol{\theta}^\star}(\mathbf{x}) \mathrm{d}\mathbf{x}}_{=1}}_{=0} - \nabla^2_{\boldsymbol{\theta}',\boldsymbol{\theta}'} \mathcal{H}_2(\boldsymbol{\theta}^\star, \boldsymbol{\theta}^\star) \tag{91}$$

$$= -\nabla^2_{\boldsymbol{\theta}',\boldsymbol{\theta}'} \mathcal{H}_2(\boldsymbol{\theta}^\star, \boldsymbol{\theta}^\star) \ . \tag{92}$$

Consequently, with $\mathbf{A} = \nabla^2_{\boldsymbol{\theta}',\boldsymbol{\theta}'} \mathcal{H}_1(\boldsymbol{\theta}^\star, \boldsymbol{\theta}^\star)$ and $\mathbf{B} = \nabla^2_{\boldsymbol{\theta},\boldsymbol{\theta}'} \mathcal{H}_2(\boldsymbol{\theta}^\star, \boldsymbol{\theta}^\star)$ we get

$$\mathcal{JG}(\boldsymbol{\theta}^\star) = (\mathbf{A} + \lambda \mathbf{B})^{-1} \cdot \lambda \mathbf{B} \tag{93}$$

$$= \left( \mathbf{I}_d + \lambda \mathbf{A}^{-1} \mathbf{B} \right)^{-1} \cdot \lambda \mathbf{A}^{-1} \mathbf{B} \ . \tag{94}$$

$\square$

*Proof.* Now, we have that

$$\|\mathbf{A}^{-1}\mathbf{B}\| = \|\mathbf{A}^{-1}(\mathbf{B} - \mathbf{A}) + \mathbf{I}_d\| \tag{95}$$

$$\leq 1 + \|\mathbf{A}^{-1}\| \cdot \|\mathbf{B} - \mathbf{A}\| \tag{96}$$

$$\leq 1 + \frac{L\varepsilon}{\alpha} \ , \tag{97}$$

where we used the triangle inequality, the submutliplicativity of the matrix norm and [Assumptions 1](#) and [2](#) yield

$$\|\mathbf{B} - \mathbf{A}\| = \left\| \nabla^2_{\boldsymbol{\theta}, \boldsymbol{\theta}'} \mathcal{H}_2(\boldsymbol{\theta}^\star, \boldsymbol{\theta}^\star) - \nabla^2_{\boldsymbol{\theta}', \boldsymbol{\theta}'} \mathcal{H}_1(\boldsymbol{\theta}^\star) \right\| \tag{98}$$

$$= \left\| \int_X \nabla^2_{\boldsymbol{\theta}, \boldsymbol{\theta}} \log p_{\boldsymbol{\theta}^\star}(\mathbf{x}) p_{\boldsymbol{\theta}^\star}(\mathbf{x}) \mathrm{d}\mathbf{x} - \int_X \nabla^2_{\boldsymbol{\theta}, \boldsymbol{\theta}} \log p_{\boldsymbol{\theta}^\star}(\mathbf{x}) p_{\mathrm{data}}(\mathbf{x}) \mathrm{d}\mathbf{x} \right\| \tag{99}$$

$$= \left\| \mathbb{E}_{\mathbf{x} \sim p_{\boldsymbol{\theta}^\star}} \left[ \nabla^2_{\boldsymbol{\theta}, \boldsymbol{\theta}} \log p_{\boldsymbol{\theta}^\star}(\mathbf{x}) \right] - \mathbb{E}_{\mathbf{x} \sim p_d} \left[ \nabla^2_{\boldsymbol{\theta}, \boldsymbol{\theta}} \log p_{\boldsymbol{\theta}^\star}(\mathbf{x}) \right] \right\| \tag{100}$$

$$\leq L d_W(p_{\boldsymbol{\theta}^\star}, p_{\mathrm{data}}) \ , \tag{101}$$

by Kantorovich-Rubenstein duality theorem, where $L$ is the Lipschitz norm of $\mathbf{x} \mapsto \nabla^2_{\boldsymbol{\theta}, \boldsymbol{\theta}} \log p_{\boldsymbol{\theta}}(\mathbf{x})$.
□

*Proof.* ([Theorem 1](#)) Using [Lemma E.1 *iii)*](#), one has that if $\lambda(1 + \varepsilon L/\alpha) < 1$, then $\|\lambda \mathbf{A}^{-1} \mathbf{B}\| < 1$ and thus

$$\left( \mathbf{I}_d + \lambda \mathbf{A}^{-1} \mathbf{B} \right)^{-1} = \sum_{k=0}^{\infty} (-\lambda \mathbf{A}^{-1} \mathbf{B})^k \ . \tag{102}$$

Combining [Lemma E.1 *ii)*](#) and [eq. (102)](#) yields

$$\mathcal{J}\mathcal{G}(\boldsymbol{\theta}^\star) = - \sum_{k=1}^{\infty} (-\lambda \mathbf{A}^{-1} \mathbf{B})^k \ . \tag{103}$$

Thus, by the triangle inequality and the submutliplicativity of the matrix norm,

$$\|\mathcal{J}\mathcal{G}(\boldsymbol{\theta}^\star)\| \leq \sum_{k=1}^{\infty} \|\lambda \mathbf{A}^{-1} \mathbf{B}\|^k = \frac{\|\lambda \mathbf{A}^{-1} \mathbf{B}\|}{1 - \|\lambda \mathbf{A}^{-1} \mathbf{B}\|} \ . \tag{104}$$

To conclude, one simply needs to provide a sufficient condition for $\frac{\|\lambda \mathbf{A}^{-1} \mathbf{B}\|}{1 - \|\lambda \mathbf{A}^{-1} \mathbf{B}\|} < 1$, which is $\lambda \left(1 + \frac{L\varepsilon}{\alpha}\right) < 1/2$.
□

# F    PROOF OF THEOREM 2

**Assumption 1.** *For $\boldsymbol{\theta}$ close enough to $\boldsymbol{\theta}^\star$, the mapping $x \mapsto \nabla_{\boldsymbol{\theta}}^2 \log p_{\boldsymbol{\theta}}(\mathbf{x})$ is L-Lipschitz.*

**Assumption 2.** *The mapping $\boldsymbol{\theta} \mapsto \mathbb{E}_{\mathbf{x} \sim p_{\text{data}}}[\log p_{\boldsymbol{\theta}}(\mathbf{x})]$ is continuously twice differentiable locally around $\boldsymbol{\theta}^\star$ and $\mathbb{E}_{\mathbf{x} \sim p_{\text{data}}}[\nabla_{\boldsymbol{\theta}}^2 \log p_{\boldsymbol{\theta}^\star}(\mathbf{x})] \preceq -\alpha \mathbf{I}_d \prec 0$.*

**Theorem 2.** *(Approximate Stability) Under the same assumptions of Theorem 1 and Assumption 3, we have that there exists $0 < \rho < 1$ and $R > 0$ such that if $\|\boldsymbol{\theta}_0^n - \boldsymbol{\theta}^\star\| \leq R$ then, with probability $1 - \delta$,*

$$\|\boldsymbol{\theta}_t^n - \boldsymbol{\theta}^\star\| \leq \left( \varepsilon_{\text{OPT}} + \frac{a}{\sqrt{n}} \sqrt{\log \frac{bt}{\delta}} \right) \sum_{l=0}^{t} \rho^l + \rho^t \|\boldsymbol{\theta}_0 - \boldsymbol{\theta}^\star\|, \quad \forall t \in \mathbb{N}^* . \tag{14}$$

**Assumption 3.** *There exists $a, b, \varepsilon_{\text{OPT}} \geq 0$ and a neighborhood $U$ of $\boldsymbol{\theta}^\star$ such that with probability $1 - \delta$ over the samplings, we have[5]*

$$\forall \boldsymbol{\theta} \in U, \forall n \in \mathbb{N}, \|\mathcal{G}_\lambda^n(\boldsymbol{\theta}) - \mathcal{G}_\lambda^\infty(\boldsymbol{\theta})\| \leq \varepsilon_{\text{OPT}} + \frac{a}{\sqrt{n}} \sqrt{\log \frac{b}{\delta}} . \tag{13}$$

*Proof.* (Theorem 2)

$$\left\|\boldsymbol{\theta}_{t+1}^n - \boldsymbol{\theta}^\star\right\| \leq \left\|\boldsymbol{\theta}_{t+1}^n - \boldsymbol{\theta}_{t+1}^\infty\right\| + \left\|\boldsymbol{\theta}_{t+1}^\infty - \boldsymbol{\theta}^\star\right\| \tag{105}$$

$$\leq \|\mathcal{G}_{\lambda,\zeta}^n(\boldsymbol{\theta}_t^n) - \mathcal{G}_\lambda(\boldsymbol{\theta}_t^n)\| + \|\mathcal{G}(\boldsymbol{\theta}_t^n) - \mathcal{G}(\boldsymbol{\theta}^\star)\| . \tag{106}$$

Using Assumption 3, with probability $1 - \delta$ we have that

$$\left\|\mathcal{G}_{\lambda,\zeta}^n(\boldsymbol{\theta}_t^n) - \mathcal{G}_\lambda(\boldsymbol{\theta}_t^n)\right\| \leq \varepsilon_{\text{OPT}} + \frac{a}{\sqrt{n}} \sqrt{\log \frac{b}{\delta}} , \tag{107}$$

Moreover Theorem 1 states that there exists a constant $\rho < 1$ such that

$$\|\mathcal{G}(\boldsymbol{\theta}_t^n) - \mathcal{G}(\boldsymbol{\theta}^\star)\| \leq \rho \|\boldsymbol{\theta}_t^n - \boldsymbol{\theta}^\star\| . \tag{108}$$

This yields that with probability $1 - \delta$

$$\left\|\boldsymbol{\theta}_{t+1}^n - \boldsymbol{\theta}^\star\right\| \leq \left( \varepsilon_{\text{OPT}} + \frac{a}{\sqrt{n}} \sqrt{\log \frac{b}{\delta}} \right) + \rho \left\|\boldsymbol{\theta}_t^n - \boldsymbol{\theta}^\star\right\| . \tag{109}$$

A recurrence and the conditional independence of the successive samplings yield

$$\forall t \in \mathbb{N}^*, \delta \in [0,1], \mathbb{P}\left( \|\boldsymbol{\theta}_t^n - \boldsymbol{\theta}^\star\| \leq \left( \varepsilon_{\text{OPT}} + \frac{a}{\sqrt{n}} \sqrt{\log \frac{b}{\delta}} \right) \sum_{i=0}^{t} \rho^i + \rho^t \|\boldsymbol{\theta}_0 - \boldsymbol{\theta}^\star\| \right) \geq (1-\delta)^t . \tag{110}$$

As done in Duchi (2009) or in Bubeck et al. (2021, Proof sketch of Prop. 2), the change of variable $\delta' = \delta \cdot t$ yields

$$\forall t \in \mathbb{N}^*, \delta' \in [0,t], \mathbb{P}\left( \|\boldsymbol{\theta}_t^n - \boldsymbol{\theta}^\star\| \leq \left( \varepsilon_{\text{OPT}} + \frac{a}{\sqrt{n}} \sqrt{\log \frac{bt}{\delta'}} \right) \sum_{i=0}^{t} \rho^i + \rho^t \|\boldsymbol{\theta}_0 - \boldsymbol{\theta}^\star\| \right) \geq \underbrace{(1 - \delta'/t)^t}_{\geq 1 - \delta'} . \tag{111}$$

In particular, since $t \geq 1$, Equation (111) implies

$$\forall t \in \mathbb{N}^*, \delta' \in [0,1], \mathbb{P}\left( \|\boldsymbol{\theta}_t^n - \boldsymbol{\theta}^\star\| \leq \left( \varepsilon_{\text{OPT}} + \frac{a}{\sqrt{n}} \sqrt{\log \frac{bt}{\delta'}} \right) \sum_{i=0}^{t} \rho^i + \rho^t \|\boldsymbol{\theta}_0 - \boldsymbol{\theta}^\star\| \right) \geq \underbrace{(1 - \delta'/t)^t}_{\geq 1 - \delta'} . \tag{112}$$

$\square$

---

[5]We could have kept a more general bound as $C(n, \delta)$, and this choice comes without loss of generality.

# G  ADDITIONAL EXPERIMENTS AND DETAILS

## G.1  EIGHT GAUSSIANS – DDPM

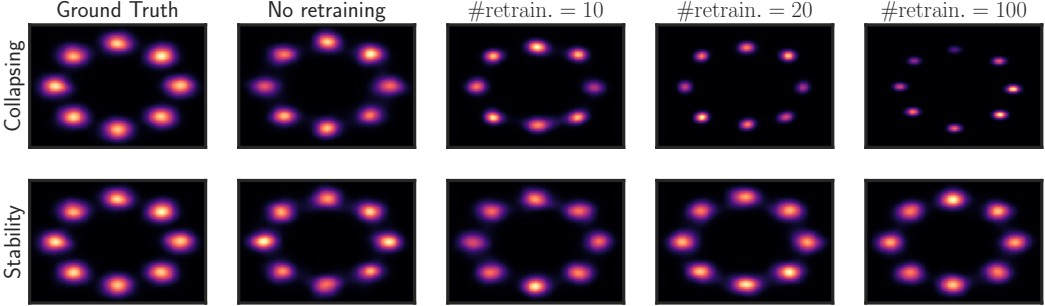

Figure 3: **Stability vs. collapsing of iterative retraining of generative models on their own data.** Each model's density is displayed as a function of the number of retraining steps. The first two columns correspond to the true density and the density of a diffusion model trained on the true data. As observed in Shumailov et al. (2023); Alemohammad et al. (2023), iteratively retraining the model exclusively on its own generated data (top row) yields a density that collapses: samples very near the mean of each mode are sampled almost exclusively after 100 iterations of retraining. Contrastingly, retraining on a mixture of true and generated data (bottom row) does not yield a collapsing density.

## G.2  TWO MOONS – FLOW MATCHING

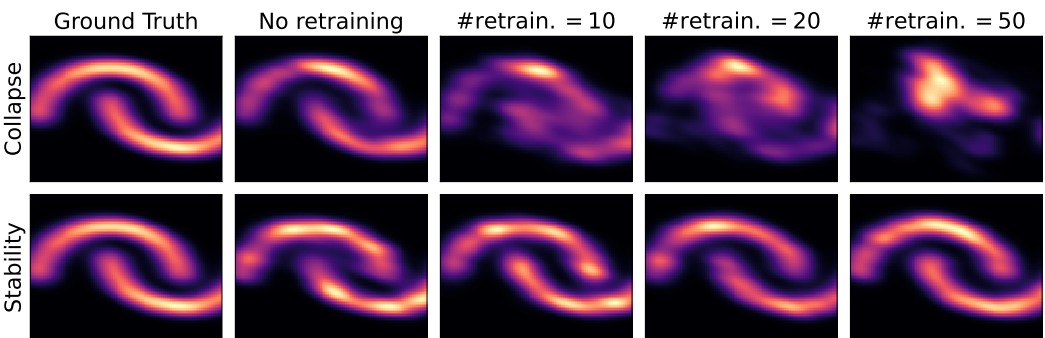

Figure 4: **Stability vs. collapsing of iterative retraining of generative models on their own data.** Each model's density is displayed as a function of the number of retraining steps. The first two columns correspond to the true density and the density of a diffusion model trained on the true data respectively.

## G.3    CIFAR-10 – DDPM

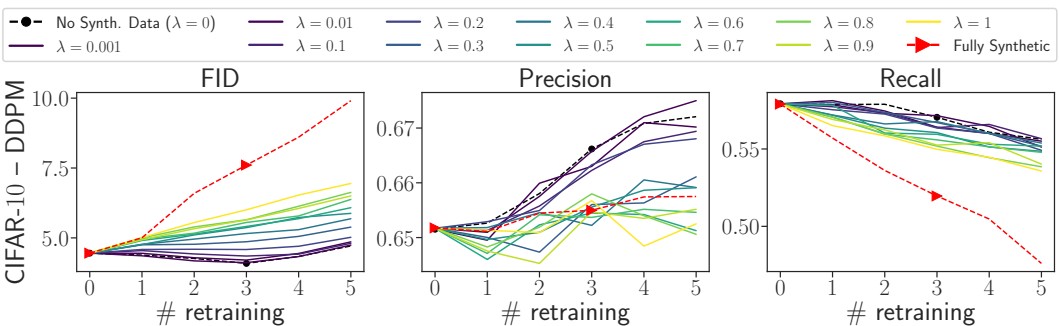

Figure 5: **FID, precision, and recall of the generative models as a function of the number of retraining for multiple fractions** $\lambda$ **of generated data,** $\mathcal{D} = \mathcal{D}_{\text{real}} \cup \{\tilde{\mathbf{x}}_i\}_{i=1}^{\lfloor \lambda \cdot n \rfloor}$, $\tilde{\mathbf{x}}_i \sim p_{\boldsymbol{\theta}_t}$. Only training on synthetic data (dashed red line with triangles) yields divergence.

For DDPM, since the optimizer was not provided in the checkpoints, we first train the model from scratch for 1000 epochs and use it as a pretrained model.

## G.4  Full Graphs for Fully Synthetic CIFAR10-OTCFM, CIFAR10-EDM, and FFHQ-EDM

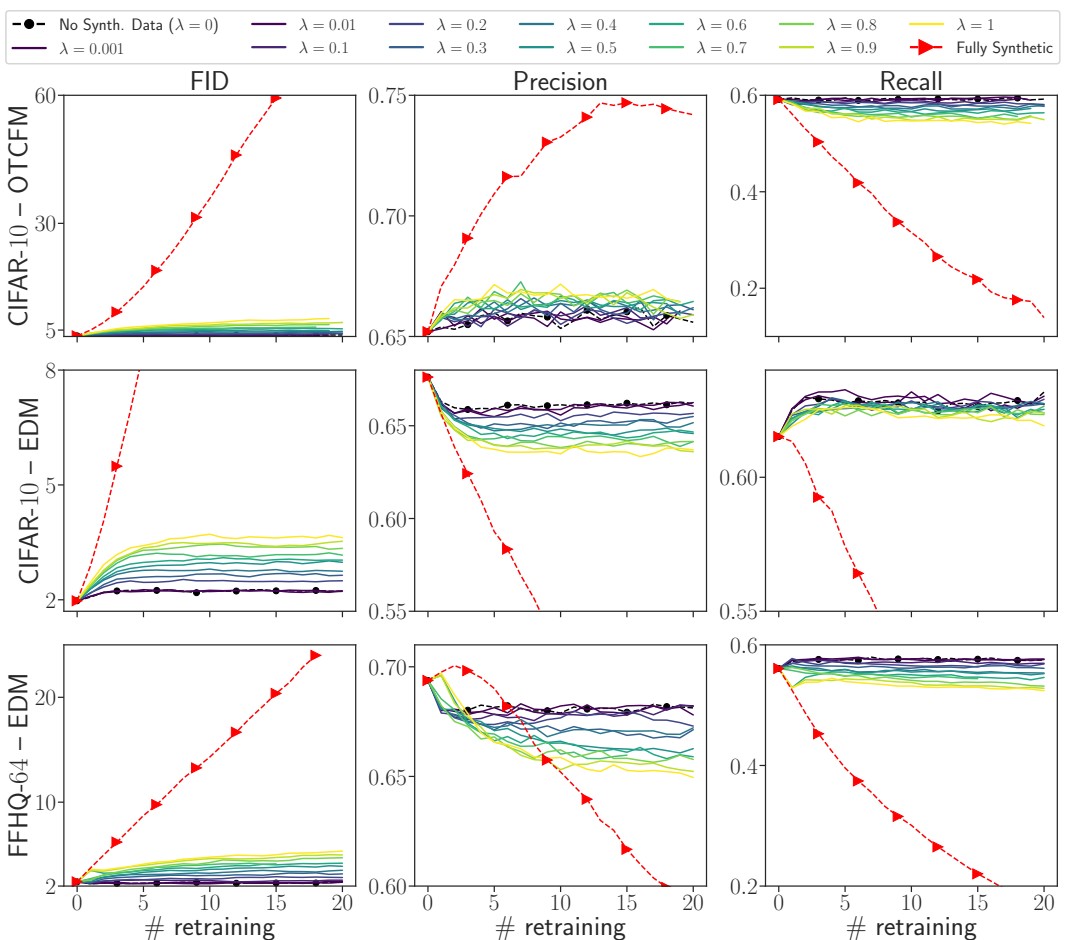

Figure 6: **FID, precision, and recall of the generative models as a function of the number of retraining for multiple fractions** $\lambda$ **of generated data,** $\mathcal{D} = \mathcal{D}_{\text{real}} \cup \{\tilde{\mathbf{x}}_i\}_{i=1}^{\lfloor \lambda \cdot n \rfloor}, \tilde{\mathbf{x}}_i \sim p_{\boldsymbol{\theta}_t}$. For all models and datasets, only training on synthetic data (dashed red line with triangles) yields divergence. For the EDM models on CIFAR-10 (middle row), the iterative retraining is stable for all the proportions of generated data from $\lambda = 0$ to $\lambda = 1$. For the EDM on FFHQ-64 (bottom row), the iterative retraining is stable if the proportion of used generated data is small enough ($\lambda < 0.5$).

## G.5  Details on the Metrics: Precision and Recall

**High Level Idea**    The notion of precision and recall used for generative models is different from the one used in standard supervised learning. It was first introduced in "Assessing Generative Models via Precision and Recall" (Sajjadi et al., 2018) and refined in "Improved precision and recall metric for assessing generative models" (Kynkäänniemi et al., 2019), which we use in practice. Intuitively, at the distribution level, precision and recall measure how the training and the generated distributions overlap. Precision can be seen as a measure of what proportion of the generated samples are contained in the "support" of the training distribution. On the other hand, recall measures what proportion of the training samples are contained in the "support" of the generated distribution. If the training distribution and the generated distribution are the same, then precision and recall should be perfect. If there is no overlap—i.e. disjoint between the training distribution and the generated distribution, then one should have zero precision and zero recall.

**Implementation Details** In order to adapt these metrics for empirical distributions that have finite samples customary of training deep generative models, multiple numerical tricks are required. Specifically, for an empirical set of samples, the support of the associated distribution is approximated by taking the union of balls centered at each point with radius determined by the distance to the k-th nearest neighbor. This is generally done in some meaningful feature space (here we use Inception V3 network, Heusel et al. 2017). For FID, we follow the standard of comparing 50k generated samples to the 50k samples in the training set. For precision and recall, with the same sets of samples, we follow the methodology of (Kynkäänniemi et al., 2019) with $k = 4$. We used the FID, precision and recall implementations of Jiralerspong et al. (2023) (`https://github.com/marcojira/fls`).

## G.6 SAMPLE VISUALIZATION

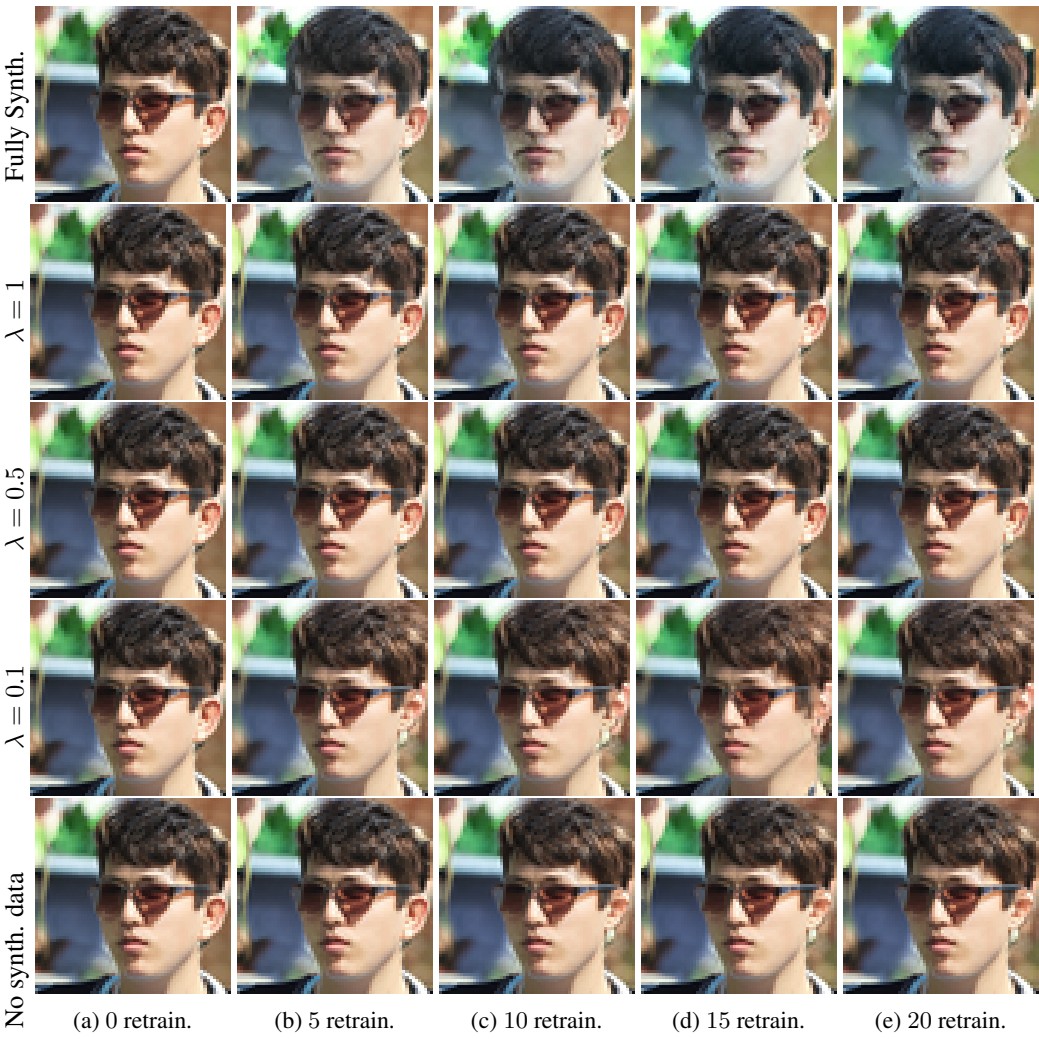

(a) 0 retrain.    (b) 5 retrain.    (c) 10 retrain.    (d) 15 retrain.    (e) 20 retrain.

Figure 7: **EDM samples, with generative models iteratively retrained on their own data, only on the synthetic data (top), and on a mix of synthetic and real data.**

### G.7 QUANTITATIVE SYNTHETIC EXPERIMENTS

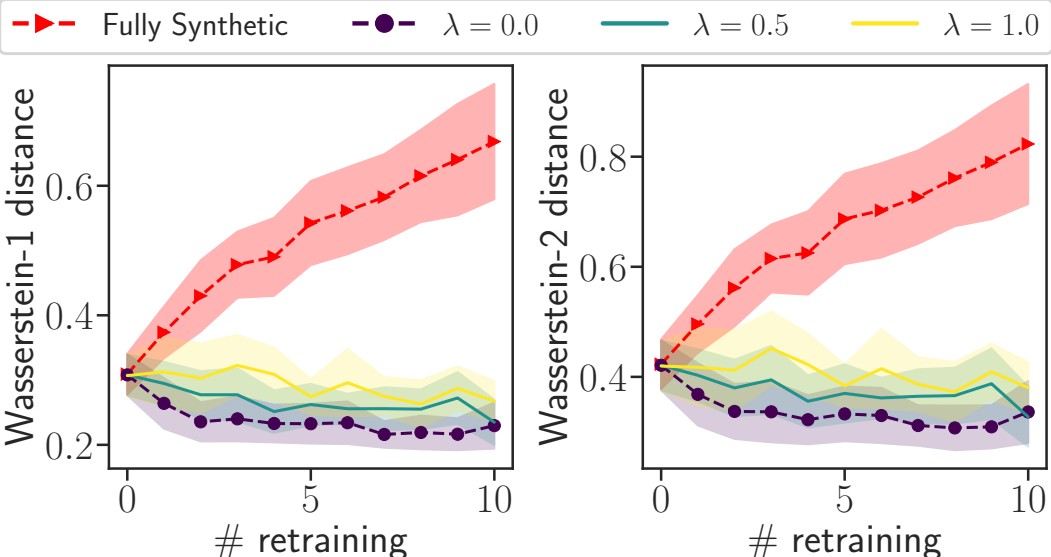

Figure 8: **Stability vs. collapsing of iterative retraining of generative models on their own data.** The Wasserstein-1 and 2 distances between the true data distribution and the generated one is displayed as a function of the number of retraining. The distances are averaged over 50 runs, the line corresponds to the mean, and the shaded area to the standard deviation. When the model is fully retrained only on its own data (Fully Synthetic, dashed red line with triangles), the distance to the true data distribution diverges. When the model is retrained on a mixture of real and synthetic data ($\lambda = 0$, $\lambda = 0.5$, $\lambda = 1$), then the distance between the generated samples and the true data distribution stabilizes.

The setting of Figure 8 is the same as for Figure 4: we used the two-moons datasets, and learn the samples using OT-CFM (Tong et al., 2023), a state-of-the art flow model, which corresponds to a continuous normalizing flow, which is an exact likelihood. The dataset has 1000 samples, and all the hyperparameters of the OT-CFM are the default ones from the implementation of Tong et al. (2023), https://github.com/atong01/conditional-flow-matching/blob/main/examples/notebooks/SF2M_2D_example.ipynb.

## G.8 PARAMETER CONVERGENCE

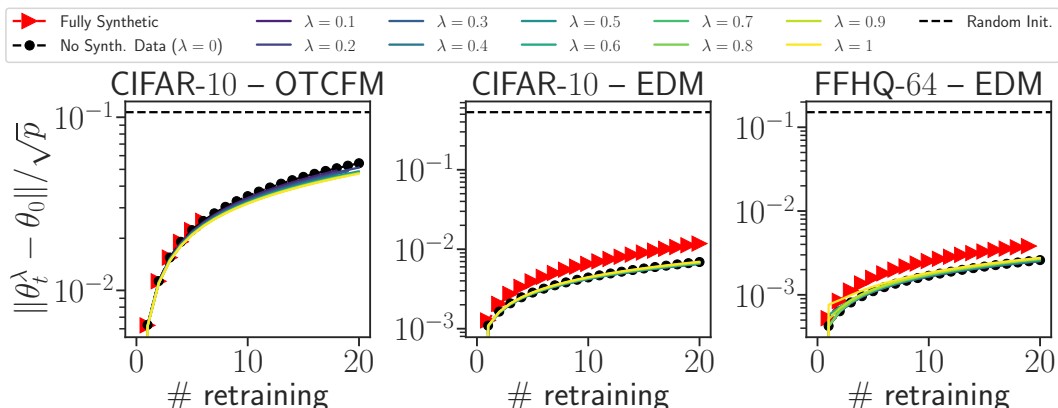

Figure 9: **Convergence in the Parameter Space.** The figure displays the Euclidean norm of the difference between the current parameters at retraining $t$ $\boldsymbol{\theta}_t^\lambda$, and the parameters $\boldsymbol{\theta}_{t_0}$ of the initial network used to finetune. In order to have comparable scales between the models and the dataset, the norm is rescaled by the number of parameters $p$ of each network. The dashed black line shows the norm of the difference between the parameter $\boldsymbol{\theta}_{t_0}$ of the initial network used to finetune, and a random initialization. Each column displays a different combinaison of dataset and algorithm. One can that fully retraining on synthetic data yields to a larger distance to the initial parameters, than retraining on a mix of synthetic and real data.

The setting of Figure 9 is the same as for Figure 2, we finetune the EDM (Karras et al., 2022) and OTCFM models (Lipman et al., 2022; Tong et al., 2023) for 20 epochs and 20 retraining on CIFAR-10 and FFHQ.

