# OpenReview forum: "On the Stability of Iterative Retraining of Generative Models on their own Data"
_ICLR.cc/2024/Conference — ICLR 2024 spotlight_

### Official Review · Reviewer_LQsW · 2023-10-27

**Soundness:** 3 good
**Presentation:** 4 excellent
**Contribution:** 4 excellent
**Rating:** 8
**Confidence:** 3

**Summary:**

This article studies the stability of iteratively training generative models with generated samples as part of the training set. In this article, it is proven that under some regulatory and optimality assumptions, the model after many rounds of training will still be close to the original one. This article considers both the infinite-sample and finite-sample cases, and provide error bounds on model distances. Then, the article studies several popular deep generative models on standard benchmarks to investigate what happens in practice.

**Strengths:**

- The iterative training formulation has become a very important problem today as there are many powerful deep generative models and their generated samples are used to train or finetune other models. This article presents a very concise and elegant way to describe this task, which is able to leverage previous theory on deep generative models by its nature.
- The assumptions are mild and the theoretical results are good. It is not surprising that with a small enough $\lambda$ the iterative training will be stable, but it is encouraging that $\lambda$ can be as large as $1/4$ in Thm 1.
- The presentation of this paper is very clean. It is very easy to follow from background and preliminaries to assumptions, theorems, and proofs.

**Weaknesses:**

The main weakness of this article is that its experiments cannot fully justify the theoretical analysis. All experiments on the high-resolution image tasks are based on diffusion models, where there might be some shift from the theoretical analysis on maximum likelihood training as diffusion models optimize variants of ELBO. There should be experiments on models trained with the exact likelihood (and on real world datasets), such as flows and autoregressive models. While flows may have a larger $\epsilon$ due to their capacity issues, autoregressive models might be a better objective to look at. I'd like to see some experiments on this.

Regarding Fig 3, the trends are not clear enough, and I'd like to see results for more iterations so that the trends become clear. The differences between different runs are very small, so the authors should run multiple experiments with different random seeds to reduce the effect caused by randomness. In addition, I'd like to see results on  $\parallel \theta_t - \theta^* \parallel$ for more direct comparison.

**Questions:**

Please refer to the weakness section for questions on experiments.


--------------

**After rebuttal** the authors have improved or added experiments that fully addressed my concerns. The results make the paper much stronger than the first draft. I think this paper is novel, sound, enlightening, and opens a new window to look at the current challenges of modern generative models.

---

> ### Author Response · Authors · 2023-11-20
> **Rebuttal Reviewer LQsW**
>
> We would like to thank the reviewer for highlighting the clarity of the manuscript, the importance of the topic, and the significance of the results. We implemented most of the propositions of Revierwer LGsW, which significantly improved the quality of the paper
>
> **Exact Likelihood Model**
>
> We would like to thank Reviewer LQsW for their insightful comments. In the new version of Figure 3, **we now include a state-of-the-art flow model**: conditional flow matching [1,2]. This corresponds to a continuous normalizing flow, which is an exact likelihood model as the reviewer requested. This model was trained on the image dataset CIFAR-10, and we observe the exact same trend with $10$ iterative retraining steps (by the end of the rebuttal we hope to update this plot with 20 steps).
>
> **More Epochs and Retraining**
>
> As stated in the global response to all reviewers, we have added more epochs (from 5 or 10 to 20) and more retraining (from 5 to 20). This yields much clearer results: one can see a stabilization after 10 to 15 retraining, especially when compared to retraining on fully synthetic data.  This is also supported by the visual inspection of the samples added in Appendix G.3 which shows almost no degradation when retraining on a mix of synthetic and generated data.
>
> **Seeds**
>
> We acknowledge the reviewer’s comment regarding the number of seeds used in our iterative retraining experiments for natural image datasets. We first highlight that we have improved this experiment in the updated version of the manuscript but substantially increased the number of iterative retraining steps ($5$ vs $20$) while adding a fully synthetic baseline ($\lambda = \infty$). **In our plots, we see that the margin of difference between the fully synthetic case and the more stable regime of $\lambda > 0$ of iterative retraining is quite dramatic**. While the reviewer has requested more random seeds, we wish to politely push back due to the high computational cost of running each experiment. For example, just for FFHQ-64, conducting iterative retraining using the SOTA EDM model amounts to a full week of experimentation using 40 RTX8000 GPUs. Unfortunately, running additional random seeds for all models and datasets is beyond the computational budget within the period of this rebuttal. In addition, as required by reviewer bfKK, we have added quantitative measurements for the experiments on the 2D synthetic data and were able to repeat this experience multiple times and provide statistics such as confidence intervals and error bars.
>
> We thank the reviewer again for their review and detailed comments that helped strengthen the paper. We believe we have answered to the best of our ability all the great questions raised by the reviewer. We hope our answer here and in the global response allows the reviewer to consider potentially upgrading their score if it they see fit. We are also more than happy to answer any further questions.
>
>
> [1] Lipman, Y., Chen, R.T., Ben-Hamu, H., Nickel, M. and Le, M., 2022. Flow matching for generative modeling. arXiv preprint arXiv:2210.02747.
>
> [2] Tong, A., Malkin, N., Huguet, G., Zhang, Y., Rector-Brooks, J., Fatras, K., Wolf, G. and Bengio, Y., 2023, July. Improving and generalizing flow-based generative models with minibatch optimal transport. In ICML Workshop on New Frontiers in Learning, Control, and Dynamical Systems.

---

> > ### Comment · Reviewer_LQsW · 2023-11-20
> > **Response to the authors**
> >
> > I thank the authors for the addition experiments. The conditional flow matching experiments make a very good complement to diffusion models. The extended x-axis on training steps shows stability after ~10 epochs. The experiments in G.4 shows variances over different random seeds and makes sense to me. These are very excellent improvements over the first submission.
> >
> > One last request from me is to include a figure just like fig. 3 but the y-axis is the direct model distances $\parallel\theta_t-\theta^*\parallel$. It is completely reasonable even if the distances turn out to be not small (for example, because of neuron permutations; in such case I recommend the authors to use techniques like git-rebasin to alleviate this issue). Nevertheless, I still think it contains very important information for us to know - as it is a direct visualization of the theorems in this paper.

---

> ### Author Response · Authors · 2023-11-21
> **Plot in the parameter space**
>
> We thank the reviewer for their warm comments on the updated manuscript.
>
> ## Plot in the parameter space
> We are grateful for the reviewer's insightful feedback and ongoing engagement in this rebuttal process.  First, we would like to say that ideally, Theorem 1 and 2 would have been proved in the distribution space, i.e., on the distribution $p_{\theta}$, not the parameters $\theta$. However, the infinite dimension of the probability space currently makes it out of reach for our current proof.
>
> To further substantiate our theory—on $\theta$—and as the reviewer rightfully encouraged—we have included a new appendix G.5 with plots of the direct model distance in parameter space for the CIFAR10 – EDM, FFHQ-64 EDM, and CIFAR-10 OTCFM experiments. In particular, we plot the parameter distance between an iteratively retrained model at timestep $t$ to the initial model $\theta_0$, $ || \theta^{\lambda}_t - \theta_0 ||_2 / \sqrt{n}$ for $t \in [0, 20]$. We further ablate the influence of $\lambda$, with $\lambda = \infty$ corresponding to the fully synthetic setting in our paper. Finally, we compare $\theta_0$ with a randomly initialized (and untrained network) to ground this study.
>
> Our results in Appendix G.5, for EDM, Figure 7 shows that when the number of iterative retraining increases, the average parameter $L_2$ distance stays in a range between $0.0005$ and $0.001$. In contrast, the randomly initialized network has a distance order of magnitude larger than $ || \theta^{\lambda}_t - \theta_0 ||_2 / \sqrt{n}$. This means a random network is extremely far away in average $L_2$ distance to $\theta_0$. These results further corroborate the results in Figure 3. The main theory of our paper is that for stable regions, the average parameter distance for iterative retraining to $\theta_0$ remains small for small values of $\lambda$.
>
> ## Git-rebasin
> We thank the reviewers for their insightful suggestions. Because of the multitude of local minima, we also think that git-rebasin-like is the way to compare the convergence in the parameter space properly. Unfortunately, the current public implementations are model-specific (e.g., specific to a ResNet 100) and do not work directly for our models. This would require significant additional time to adapt git re-basin for our architectures.

---

> > ### Comment · Reviewer_LQsW · 2023-11-21
> > **Reply to the authors**
> >
> > I thank the authors for the additional experiments.
> >
> > The statement that distances should be measured in the distribution space makes sense to me, and I hope the authors could make that more explicit in the paper. The numerical results do show a trend that smaller $\lambda$ leads to smaller model distances, which is more obvious in the third figure. This is very useful information. I hope the authors can improve the visualization of these plots and make the lines easier to distinguish in their final version.

---

> ### Author Response · Authors · 2023-11-21
> **Kindly waiting for more feedback**
>
> Thank you again for your suggestions and prompt response. We hope that the additional Figure (Fig. 7 in Appendix G.5) provides the desired level of clarity and information for the convergence in the parameter space, and convincingly answers any lingering doubts. We appreciate your time and effort in this rebuttal period and hope that our answers are exhaustive enough to convince you to upgrade your score.

---

### Official Review · Reviewer_9HHH · 2023-10-30

**Soundness:** 2 fair
**Presentation:** 3 good
**Contribution:** 2 fair
**Rating:** 5
**Confidence:** 3

**Summary:**

The authors study the problem of generative models that are recursively trained with parts of the dataset are generated by an earlier version of the generative model. They provide theoretical proofs that suggest that if the share of generated data w.r.t. the original data is small enough, this training can still be stable, whereas otherwise it collapses.

**Strengths:**

**Well-structured.** The structure of the paper seems fine and I like how the simple cases are considered first to build an intuition before more complex cases are presented. Although the matter of the paper is quite technical, the formal introduction of the problem and the notation is well executed. Apart from one point (see "clarity" below), I think the overall presentation is good.

**Interesting and Timely problem.** the problem studied in this work is interesting and can be considered timely as indeed, content by generative AI is flooding the internet, which in turn is the key source of training data for generative models.

**Technical rigor.** I have the overall impression that the technical part is rigorously executed and I did not find any significant flaws. However, as pointed out below, I was not able to verify all steps of the proofs.

**Weaknesses:**

**Clarity.** I am a bit puzzled by the use of $\succcurlyeq$ for matrices. When you write $\Sigma \succcurlyeq 0$, I suppose it is the usual condition for Sigma to be positive definite and not that each individual element of sigma should be larger than zero. This would rule out certain covariance matrices. On the other hand, the authors write $\nabla^2 f = H_f \preccurlyeq -\alpha I_d$, comparing two matrices (supposing that $I_d$ is the $d$-dimensional unit matrix). I suppose that it now constrains each element. Can the authors please clarify?

Measures: What is meant by precision and recall in the evaluation section? Is there a discriminator deployed that tries to differentiate between the real and fake samples? This needs to be clarified. I don’t currently see how generative models can have recall/precision.

**Clarity of the Proofs.** Unfortunately, some proofs in the paper offer room for improvement of clarity and were inaccessible for verification in their current form. For instance:
In appendix A.1., I can follow the proof of Lemma 1 and Lemma 2. However, it is not obvious how equation (13) implies the form of $\alpha(n)$ using the $\Gamma$-function. This should necessarily be clarified as there is not a word on how the form of $\alpha(n)$ comes into play.

**Some assumptions made in this work may not reflect reality well and may be oversimplifying.** While I know that certain assumptions are required to make the problem amenable to theoretical analysis, I have concerns that they may be overly restrictive in practice. In particular, we basically assume convexity of the loss function through assumption 1 and 2. As far as I can see, due to the Lipschitz constant on the Hessian being L (assumption 1) and the Hessian's eigenvalues being smaller than $-\alpha$ (interpreting the operator that way, please correct me in case this is not correct), in the ball of $\epsilon = \alpha/L$ around $\theta^*$, the Hessian will be negative definite, implying convexity. As the Theorems only state the existence of a radius $\delta > 0$ in which the convergence properties hold, we basically consider the convex part of the problem. As we know, modern loss landscapes are far from convex. It is highly unlikely that both the initial value and the optimum $\theta^*$ lie in a ball in which the loss function is convex.
Another impractical assumption may be the assumption of no approximation error. This is usually only shown for infinite-parameter models. I would be fine with these approximations if the empirical results would confirm the assumptions and the analysis that follows from them. However the evaluation results seem to rather confirm the doubts.

**The evaluation does not confirm the claims.** As far as I can tell, the point of departure for this work is as follows: Previous works [1,2] have already established that model training on solely generated data is unstable or collapses. On the other hand, training models again and again on the same dataset is stable (otherwise the models we currently have wouldn’t work at all, as they are trained epoch by epoch with the same data). The key claim of this work is therefore that there is some value $\lambda > 0$, i.e., a certain amount of generated data can be injected, such that model training remains stable.

Unfortunately, the experimental results are not very convincing in that regard. Indeed, the leftmost column of Figure 3 shows that the resulting FID curve is almost a linear interpolation between $\lambda=0$ (stable) and $\lambda=1$ (unstable). This suggest that for every $\lambda>0$ training will be diverge in the end. Even for the smaller lambdas, we see that the training FID gradually increases. For \lambda=0.001, I guess one would require more than 5 steps of training to observe a statistically significant effect (as we are discussion the case of infinite retraining). Furthermore, there are no measures of disparity (e.g., standard deviations) displayed, would could solidify the empirical evaluation.

**Minor points**

Related work: I am aware of the fact that the studied problem is different from mode collapse in generative models, but I have the impression that there seem to be some connections. Maybe the authors can add a discussion on this point.

Write-up: Missing parenthesis in the last line of before the statement of the contributions “(DDPM…”

Last point of the contribution section: “Using two powerful diffusion models in DDPM and EDM” (“in” seems unexpected at this place)
There are many unclarities in the proofs:

“Since most methods are local search method[s]” (use the plural form here)

Proof in Appendix C. There are some formatting errors below eqn. 33. (theta is not properly displayed).

Proof in Appendix D. Equation (39) theta’ is multiply defined, first by the outer quantor, then below the max after the $\leq$ sign. Consider using $\theta’’$ or similar in this case.


The PDF seems to render very slowly. Maybe the authors can check some of the vector graphics again to increase the overall accessibility.

**Summary:** Overall, this is an interesting work. While I do not contest the main results, I was not able to verify all proofs either as I wasn’t able to follow the arguments at some points. Furthermore, the empirical evaluation is almost contradictory to the theoretical claims in this paper. I will be willing to increase my rating to an accept-score, if the authors can clarify their proofs such that the validity of their results can be easily verified and convincingly show that values of $\lambda>0$ exist, where stable retraining is possible for a larger amount of retraining iterations (5 iterations are insufficient when considering an infinite regime).


-----------------------------

**References**

[1] S. Alemohammad, J. Casco-Rodriguez, L. Luzi, A. I. Humayun, H. Babaei, D. LeJeune, A. Siahkoohi, and R. G. Baraniuk. Self-consuming generative models go mad, 2023.

[2] I. Shumailov, Z. Shumaylov, Y. Zhao, Y. Gal, N. Papernot, and R. Anderson. The curse of recursion: Training on generated data makes models forget, 2023.

**Questions:**

1. Have the authors tried running the experiment for more than 5 steps?

2. Can the authors give standard deviations for the plots in Figure 3?

3. What do the recall and precision metrics in Figure 3 mean?

4. Can the authors clarify the >-operator for matrices?

---

> ### Author Response · Authors · 2023-11-20
> **Rebuttal Reviewer 9HHH**
>
> We would like to thank the reviewer for their time and constructive feedback. We appreciate the fact that the reviewer felt our paper was “well structured” in how simple cases are presented first to build intuition before more complex and practical examples. Moreover, we are encouraged to hear that the reviewer considers the topic of our paper an “interesting and timely problem”. Finally, we thank the reviewer for highlighting the “technical rigor” of our work. We now address the key clarification points raised by the reviewer.
>
>
> **Improving the clarity of notation**:
>
> We acknowledge the reviewer’s comments regarding the comprehensibility of our mathematical notation which we now clarify.
> $\succeq$ operator for matrices “I suppose it is the usual condition for $\Sigma$ to be positive definite and not that each individual element of sigma should be larger than zero.” Yes, this is exactly what we mean, positivity in the matrix, sense, i.e., $A \geq 0$ if and only if for all $x$, $x^\top A x \succeq 0$. In other words, the matrix $A$ is positive semidefinite.
> “On the other hand, the authors write $H_f \leq - \alpha I_d$ , comparing two matrices. suppose that it now constrains each element. Can the authors please clarify?” This choice of notation here does not constrain each element, but rather $\leq$ in the matrix sense, i.e., $A \preceq B$ if and only if $A - B \preceq 0$. This notation is sometimes referred to as Loewner order (https://en.wikipedia.org/wiki/Loewner_order),  the reader can also refer to Example 2.15 of Section 2.4.1 of the textbook [1] for more details. We believed this was conventional notation when working with matrices, but understand how this was not immediately obvious. This clarification has been added in the notation section to improve readability.
>
> **Description of Precision and Recall**
>
> We answer this question in detail in our global response to all reviewers. In summary, the precision/recall we use here differs from those used for evaluating a classification model. Intuitively, at the distribution level, precision and recall measure how the training and the generated distributions overlap. Precision can be seen as a measure of what proportion of the generated samples are contained in the “support” of the training distribution. On the other hand, recall measures what proportion of the training samples are contained in the “support” of the generated distribution.  If the training distribution and the generated distribution are the same, then precision and recall should be perfect. If there is no overlap—i.e. disjoint between the training distribution and the generated distribution, then one should have zero precision and zero recall More details and references are provided in the joint answer to reviewers and in Appendix G.2. Feel free to ask any clarification questions if needed.
>
> **Improving the clarity of the proofs.**
> We agree with the reviewer that some of our proofs could benefit from additional clarity to ease the verification of their claims. To improve readability the proofs in contention as highlighted in the review have been encapsulated into simpler lemmas, with additional intermediate steps, which we believe are much easier to follow. The key changes with respect to the initial submission have been highlighted with blue text in the updated PDF.
> More precisely, **Lemma A.1 and Lemma A.2 in Appendix A.1 and A.2 should now provide detailed guidance on the proofs** in the Gaussian case. We thank the reviewer again for helping us strengthen this aspect of our submission and are we more than happy to add anything else to these proofs should the reviewer deem it necessary.
> [1] Boyd, S.P. and Vandenberghe, L., 2004. Convex optimization. Cambridge University Press.
>
> [2] M. S.-M. Sajjadi, O. Bachem, M. Lucic, O. Bousquet, and S. Gelly. Assessing generative models via precision and recall. Neurips 2018
>
> [3] Kynkäänniemi, Tuomas, et al. "Improved precision and recall metric for assessing generative models." Advances in Neural Information Processing Systems 32 (2019).

---

> ### Author Response · Authors · 2023-11-20
> **Experimental Validation**
>
> **Experimental Validation**
>
> We concur with the reviewer that our experimental validation of our theoretical results can be improved, especially since stability is guaranteed in the infinite iterative training regime. To more convincingly demonstrate this claim we have updated our previous Figure 3 to go from $5 \to 20$ iterative retraining steps and **more epochs per retraining**, **from  $5/10 \to 20$** for the SOTA EDM model on CIFAR and FFHQ. In addition, we have also added another run to this Fig 3 plot which uses $\textit{only synthetic}$---i.e. $\lambda=0$ samples. This allows us to see a very clear signal where we can now experimentally see a clear convergence towards a stable regime in ALL metrics (FID, Precision, Recall). This is in stark contrast to the fully synthetic line in the Fig 3 plot which exhibits a linear rate of degradation. We hope this update to Fig 3. helps alleviate the reviewer's concern regarding the extrapolation behavior for $\lambda > 0$.
>
> We value the reviewer's opinion that providing a quantification of uncertainty in the vein of standard deviation would improve the empirical caliber of the results. Unfortunately, for the image experiments, this is computationally challenging. For example, the bottom row of Figure 3 (FFHQ – EDM with 20 fine-tuning epochs and 20 retraining) takes one full week of training on 40 RTX8000 GPUs. Running this experiment multiple times and for multiple models and datasets is currently beyond our computational budget given the stringent time constraints of the rebuttal period. However, we still wish to take the reviewer's suggestions into consideration and we have updated our synthetic experiments. Specifically, we included quantitative metrics such as the Wasserstein distance between the generated samples and samples from $\hat{p}_{\text{data}}$ for multiple values of $\lambda$ and also corresponding standard deviations in this setting. We hope this allows the reviewer to be more convinced by our overall experiments and how they substantiate our theory.
>
> **Over Simplifying Assumptions**
>
> Assumptions 1 and 2 (local Lipschitzness + strong convexity)  “We basically assume convexity of the loss function through assumptions 1 and 2”. We would like to highlight an important point: assumptions 1 and 2 imply local Lipschiptness and local strong convexity, which is much weaker than global Lipschitness and strong convexity, which would be indeed much stronger and unrealistic. To the best of our knowledge, local Lipschitness and strong convexity are among the mildest assumptions required for understanding the local landscape of neural networks in terms of optimization [1,3] or generalization [2]. We would be eager to consider other sets of assumptions if the reviewer can point us toward such works.
> The question on Assumptions 3 (generalization error) is answered as a joint response to reviewers
>
>
> [1] Nesterov, Y., & Polyak, B. T. Cubic regularization of Newton method and its global performance. Mathematical Programming, 2006.
>
> [2] Keskar, N.et al. On large-batch training for deep learning: Generalization gap and sharp minima. ICLR, 2017
>
> [3] Jin, Chi, et al. "How to escape saddle points efficiently." ICML. PMLR, 2017.
>
> **Minor points**
>
> We are grateful to the reviewer for pointing out the small typos and mistakes in our manuscript. We have fixed this in our updated rebuttal PDF. The PDF size of the figures has been reduced, and the PDF should render smoothly now.
>
> We thank the reviewer again for their valuable feedback and great questions. We hope that our rebuttal addresses their questions and concerns and we kindly ask the reviewer to consider upgrading their score if the reviewer is satisfied with our responses. We are also more than happy to answer any further questions that arise.

---

> ### Author Response · Authors · 2023-11-21
> **Kindly waiting for more feedback**
>
> Thank you again for your detailed review and comments. We believe we have implemented all of your great suggestions in the revised manuscript. In particular,
> - The notation and proofs should now be more straightforward (Page 3, 13, and 14),
> - Figure 3 displays now **more epochs and more retraining**, which should be more compelling, clearly illustrating Theorems 1 and 2.
> - Precision and recall are now carefully explained.
> - Details and comments on each assumption have been provided (Assumption 3 in the global response and assumptions 1,2 in the individual answer).
>
> As the end of the discussion period is fast approaching, we would love the opportunity to engage with you if there are any remaining questions (in particular, concerning the clarity of our proofs and assumptions). Otherwise, we politely encourage you to consider upgrading your score. Thank you again for your time and energy in reviewing our paper.

---

> ### Comment · Reviewer_9HHH · 2023-11-22
> **Response to Rebuttal**
>
> I thank the authors for their detailed response. Here are detailed follow-ups to the points raised:
>
> **Clarity:** In thank the authors for improving the clarity of the theory part in the revision and I increased my ratings for Presentation and Soundness. In particular, I now see how the operator $\succeq$ is to be interpreted. I also understandand how the precision / recall metric works. I checked the Proofs of Lemma A.1. and Lemma A.2 again and found that they are better to follow now. I still think that it could be more explicitly stated that the expression involving the $\Gamma$-function in Eqn. (26) comes from the analytical solution of the mean of the chi distribution, but the proof seems solid now overall. I am still mildly confused why there is an $\mathbb{E}$ around $\Sigma_0$ in the theorem. Is $\Sigma_0$ initialized randomly? As I read Proposition 1, $\Sigma_0$ is an arbitrary, but constant value?
>
> **Assumptions:** As pointed out by the authors, the theoretical analysis in this paper is concerned with strongly convex regions of the loss landscape. I think this should be explicitly stated in the paper, as the Assumptions 1+2 appear very technical, but their main effect seems to be the constraint of convexity in the region around the optimum, that is considered. Do the authors have any evidence that it is possible to find an initialization in the convex region around the optimum in practice?
>
> **Experimental Evaluation:** Thank you for updating the experiment to feature 20 retraining steps (which is still far from the infite regime but should still help to solidify the evaluation). The plot of the fully-synthetic regime also helps.
> However, my main concern still subsists. The experiments seem to show a smooth interpolation between the divergent regime with all-synthetic data and the stable case with no synthetic data. Nevertheless, while training stays stable for more iterations when more of the original data is re-used, the *infinite* regime is considered in the paper. This could be similar to adding two sequences, a divergent (that can even be multiplied with a very small $\lambda$) and a non-divergent one, which in summary would still be divergent. To me, the empirical evaluations in Fig. 3 still don't strongly support the theoretical claims of the paper that there is some value for $\lambda > 0$, for which training will also be stable in the *infinite regime*. What are the authors' thoughts on this?

---

> > ### Author Response · Authors · 2023-11-22
> >
> > We thank the reviewer for their constructive comments and continued participation in this discussion period.
> >
> > - **Clarity of the proofs**. We appreciate the reviewer’s suggestions. Indeed $\Sigma_0$ and $\mu_0$ are deterministic, so the expectation has been removed. The proof in the one-dimensional case has been simplified further. Specifically, we have removed the need to introduce Gamma functions explicitly. We hope this answers your questions on clarity.
> >
> > - **Assumptions 2 and 3.** Thank you for your suggestions. More details on the intuitions and the implications of Assumptions 1 and 2 have been added in the manuscript, especially the discussion on local strong convexity and local Lipschitzness.
> >
> > - **Initialization in a Convex Region.**
> > We believe that initializing in a convex region is possible because of:
> > 1- For modern architectures (typically, ResNet with skip connections), local optima with good generalization properties are believed to be locally convex, see for instance Figure 1-6 of *Visualizing the loss landscape of neural nets* [1], that shows that modern architectures are believed to have flat local minima.
> >  2- Our proofs and experiments significantly rely on **finetuning**. In other words, we always used **pre-trained** models. For $\lambda=0$, it is realistic to assume that the parameters of the finetuned model $\theta_0$ are already in the local convex region. Combined with the Lipschitness of the neural networks [2], it seems reasonable to assume that for good enough models and a small enough proportion of synthetic data $\lambda$, the pre-trained model's initialization parameters are in a locally convex region.
> >
> > We hope this convinces you that the initialization is possible in a locally convex region.
> >
> >
> > **Experimental Evaluation**
> >
> > We acknowledge the reviewer's concern about the empirical evaluation in Fig. 3 concerning our presented theory. We would like to politely push back against the reviewer's assertion that in the infinite regime, one could conceive of a scenario that is equivalent to adding two sequences—one divergent and one non-divergent—which may lead to instability. We politely disagree. Such a scenario is not realizable in our setup because every iterative retraining iteration uses the **same fixed $\lambda$**. Even in the infinite regime, this cannot be a superposition of two sequences of iterates with **different $\lambda$** values. Moreover, we have a complete understanding of what each iteration consists of—we simply maximize the likelihood of the (mixed) dataset—and the maximum likelihood framework is well understood for the generative models we consider in this paper (e.g., flows and diffusion). Thus, from a theoretical point of view, maximizing the likelihood corresponds to only $1$ sequence of iterates.
> >
> > We believe our current experimental findings in Fig 3 clearly show stability for many $\lambda > 0 $ values. In some instances, we even see **subtle improvements** in precision and recall see for instance the EDM graphs for small values of $\lambda$, or experiments on synthetic data (Figure 6 in Appendix G.4). Furthermore, we would like to add the paper [3] mentioned by the reviewer considered only $10$ iterations. In contrast, our paper has double the amount with $20$ iterations, and we argue the trend (of stability in our case) is equally clear. We see the slope of the lines in Figure 3 very specifically. For the stable regimes to be flat and consistently flat. Could the reviewer please clarify which parts of the plots in Fig 3 they feel do not support well the theory in this paper?
> >
> >
> >
> > [1] Li, H., Xu, Z., Taylor, G., Studer, C. and Goldstein, T., 2018. Visualizing the loss landscape of neural nets. Advances in neural information processing systems
> >
> > [2] Bubeck, S. and Sellke, M., 2021. A universal law of robustness via isoperimetry. Advances in Neural Information Processing Systems
> >
> > [3] S. Alemohammad, J. Casco-Rodriguez, L. Luzi, A. I. Humayun, H. Babaei, D. LeJeune, A. Siahkoohi, and R. G. Baraniuk. Self-consuming generative models go mad, 2023.

---

> ### Comment · Reviewer_9HHH · 2023-11-23
> **Thank you.**
>
> Thanks for the clarifications. I think the proof of Proposition 1 has been solidified now. In case of acceptance, I encourage the authors to carefully go through all proofs again, check their accessiblity and potentially add necessary intermediate steps.
>
> The results in Figure 3 appear like this to me: Suppose there is a divergent sequence $(a_n), {n \in \mathbb{N}}$ and a convergent sequence $(b_n), {n \in \mathbb{N}}$. We can now generate linear interpolations between these sequences as $c_n := \lambda a_n + (1-\lambda)b_n$. For every $\lambda > 0$ the resulting sequence will diverge.
>
> Looking at the plots, I have the impression that something similar is happening there. I would have expected a regime change for some $\lambda$, where training then becomes stable. However, it now seems like we are just interpolating between the regimes, which, in the sequence example, would always result in unstability. Although I am fully aware that the example is more complex here, I am not entirely convinced that something similar to the interpolation between sequences would not be happening if we perform more retraining iterations of the generative models. In the sequence example, we could also choose a very small $\lambda$, and divergence would not be visible after 20 steps.
>
> Overall, I think the paper's theory part is sound now, but relies on some strong assumptions which are (1) everything on the training path from initialization to the optimum being contained in the convex region around the optimum, and (2) the models being perfectly able to fit the distribution (i.e., being universal function approximators). The empirical results in the main paper have not entirely convinced me that these assumptions hold in practice and the convergence trend is hard to see due to statistical noise in these results, which unfortunately feature no measures of dispersity (e.g., std. errors).
>
> If these assumptions were clearly conveyed in the paper, I suppose we could leave it up to the reader to judge whether the theory applies to their model or not. As this has been done now, I am still not championing this paper but I have no major reservations either.

---

### Official Review · Reviewer_bfKK · 2023-10-31

**Soundness:** 3 good
**Presentation:** 4 excellent
**Contribution:** 3 good
**Rating:** 8
**Confidence:** 4

**Summary:**

In this work authors focus on the problem of retraining the generative models with the combination of real and synthesised data coming from the previous state of the same model. With a series of theoretical derivations authors show that such a process is stable (under some assumptions). The theoretical theorems are also evaluated with additional empirical studies that seem to align with the main claims.

**Strengths:**

- To my knowledge this is the first work to study the theoretical stability of generative models when retrained on its own data. It is an interesting practical problem as soon we might struggle with distinguishing true training examples from fake synthetic images. This problem is well motivated in this submission.
- In the submission authors  provide theorems with proofs that shed a new light into the topic of continual retraining of generative models with self-generated data, showing that this process might be stable under the assumption of retraining the model with sufficient share of true data samples.
- This work might have some impact on theoretical fields of ML such as continual learning of generative models and practical aspects such as deployment of big diffusion models.
- This is one of the best written theoretical papers I have ever read. Everything is extremely clear and easy to follow. It reads as a good crime!

**Weaknesses:**

- The contribution of the theoretical part has limited significance as it mostly concerns unfeasible setups with normally hard or impossible to achieve assumptions  (e.g. an infinite number of rehearsal samples generated by the model).
- The empirical evaluation of two simpler models is limited to 2 visualisations without quantitative measurements
- For Diffusion models, the evaluation is performed on 3 datasets, which is sufficient. However if I understand the setup correctly the whole analysis is performed on a single training seed. The differences presented in the plots are extremely small so it is unclear whether they are statistically significant, and therefore whether the main claims hold.

Small not important detail:
Page 8 the end of Experimental Setup section, I spend some time trying to understand what emphconstant means :D - please correct a typo

**Questions:**

Did you consider evaluation how other sampling procedures that minimises the statistical approximation error might influence the findings presented in this paper? Maybe instead of drawing random samples, but for example those that best cover the data distribution (e.g. using herding alogrithm) could prevent the model from collapsing, or at least slow it down when retrained with bigger portion of sampled data?

---

> ### Author Response · Authors · 2023-11-20
> **Rebuttal Reviewer bfKK**
>
> We thank the reviewer for their time and positive appraisal of our work. We are thrilled that the reviewer viewed our work as “one of the best-written theoretical papers I’ve ever read”. Moreover, we are glad that the reviewer finds our paper “well-motivated”, and tackles an “interesting problem” in which our paper “sheds a new light into the topic of continual retraining of generative models”. We wholeheartedly agree with the reviewer that our investigation in this paper has the potential to “impact” neighboring fields such as the continual learning of generative models and also tangibly impact the practical aspects of deploying large generative models. We now address the main questions raised by the reviewer.
>
> **Assumptions used in theory** “setups with normally hard or impossible to achieve assumptions (e.g. an infinite number of rehearsal samples generated by the model”
> We acknowledge that the infinite number of rehearsal samples is infeasible in practice,, however, we would like to point out that we provide results with a finite number of data (Theorem 2), provided that the generative models learn “well enough” (Assumption 3) the true distribution.
>
> **Empirical evaluation of simpler models**
> Quantitative metrics have been added for the synthetic experiments (see Figure 6 in Appendix G.4).  The Wasserstein-$1$ and $2$ distances between the true data distribution and the generated one is displayed as a function of the number of retraining. The distances are averaged over $50$ runs, the line corresponds to the mean, and the shaded area to the standard deviation. When the model is fully retrained only on its own data (Fully Synthetic, dashed red line with triangles), the distance to the true data distribution diverges. When the model is retrained on a mixture of real and synthetic data ($\lambda=0$, $\lambda=0.5$, $\lambda=1$), then distance between the generated samples and the true data distribution converges.
>
>
> **Evaluation over seeds**
> We acknowledge the reviewer's comment regarding the number of seeds used in our iterative retraining experiments for natural image datasets. We first highlight that we have improved this experiment in the updated version of the manuscript but substantially increased the number of iterative retraining steps (5 vs 20) while adding a fully synthetic baseline ($\lambda = \infty$). In our plots, we see that the margin of difference between the fully synthetic case and the more stable regime of $\lambda > 0$ of iterative retraining is quite dramatic. While the reviewer has requested more random seeds, we wish to politely push back due to the high computational cost of running each experiment. For example, just for one curve of FFHQ-64, conducting iterative retraining using the SOTA EDM model amounts to a full week of experimentation using 4 RTW8000 GPUs. This yields to $40$ GPUs over one week for Figure 3 itself. Unfortunately, running additional random seeds for all models and datasets is beyond the computational budget within the period of this rebuttal, and even beyond our academic resources. However, to promote reproducible research we commit to releasing the full training code of all our experiments such that one can readily verify the stability and collapse property of iterative retraining.

---

> ### Author Response · Authors · 2023-11-20
> **Other Sampling Procedures?**
>
> We are thinking of investigating other sampling procedures for the generated samples for future work. A way to do it would be to score each sample, using for instance reinforcement learning techniques. It has been explored heuristically and provides improvement of the generative models on language tasks [1]. One other way to score generated samples could be to do conditional generation, use a classifier, and look at the last layer (after the softmax). This would give a score to each sample. Then, one can each filter the examples or weigh them according to their score, as done in very recent works [2]. As mentioned by the reviewer, an additional challenge in this task is to select diverse samples, which are not copied from the training set. To the best of our knowledge, this is currently an active area of research, and are currently investigating it.
>
> This is an excellent question! There are numerous possible ways to change the inference procedure for generative models such as diffusion and flow-matching. Common examples include classifier-based and classifier-free guidance that achieves low-temperature sampling to enhance sample fidelity. More generally, this type of sampling falls into the larger goal of conditional generation which is beyond the scope of our presented theory as we are concerned with stability with respect to $p_{\text{data}}$. Another exciting direction for sampling is to score each sample, using for instance reinforcement learning techniques. This has been explored in the context of language modeling tasks [1] and has shown tangible improvements in domain-specific metrics. This direction and developing appropriate theory remain exciting directions for future research and as such are not contained within our iterative retraining framework.
>
>
> We thank the reviewer again for their time and valuable feedback. We hope our rebuttal succeeded in addressing all the salient points of the review and we kindly request the reviewer to consider upgrading their score if they so deem it. We are also available to answer any further questions that the reviewer may have.
>
> [1] Gulcehre, C., Paine, T.L., Srinivasan, S., Konyushkova, K., Weerts, L., Sharma, A., Siddhant, A., Ahern, A., Wang, M., Gu, C. and Macherey, W., 2023. Reinforced self-training (rest) for language modeling. arXiv preprint arXiv:2308.08998.
>
> [2] Hemmat, R.A., Pezeshki, M., Bordes, F., Drozdzal, M. and Romero-Soriano, A., 2023. Feedback-guided Data Synthesis for Imbalanced Classification. arXiv preprint arXiv:2310.00158.

---

> ### Author Response · Authors · 2023-11-21
> **Kindly waiting for more feeback**
>
> We would like to thank the reviewer again for their enthusiastic and constructive comments, which gave us the opportunity to improve our work significantly. In particular, **quantitative** synthetic experiments with **confidence intervals** have been added in Figure 6, in Appendix G.4, corroborating Theorems 1 and 2. Figure 3 displays now more epochs and more retraining, which showcases **stronger evidence** of stability.

---

> > ### Comment · Reviewer_bfKK · 2023-11-22
> > **Response to the authors**
> >
> > Thank you for your response and additional experiments.
> > - I appreciate additional quantitative metrics for simpler datasets
> > - I understand the limitations regarding retraining diffusion models with several seeds, but still I think this is the biggest weakness of this work especially taking into account it's rather theoretical approach.
> > - It is interesting to see the experiments with additional datasets, although (as a small not important detail) now, Figure 3 is visually unpleasant (labels are not aligned and those on the left are simply too big), it is hard to understand (with small differences), I also don't know where the Fully synthetic line goes.
> >
> > I already voted for the acceptance of this work, I still find it interesting and important, so I will keep my initial score.

---

### Official Review · Reviewer_pzxF · 2023-11-02

**Soundness:** 3 good
**Presentation:** 4 excellent
**Contribution:** 3 good
**Rating:** 6
**Confidence:** 3

**Summary:**

The authors propose an analysis of iterative retraining of generative models on their own data. This is indeed a very tempting approach used in various fields.

The analysis begins in a simple Gaussian case that enables the authors to provide intuition on the behavior of such a training method.

The analysis then addresses two other cases: under no statistical error assumption were under inifinite sampling the stability and convergence can be retrieved.

**Strengths:**

The paper is clearly presented, the objectives are well stated and i believe that the authors propositions and lemma clearly answers their problematic.

The experiments seem conclusive.

**Weaknesses:**

see questions

**Questions:**

My overall understanding of the field is limited. However i have a couple of questions:

1. How likely is assumption 3 ? can the authors provide example where such a bound stands ?
2. How is precision and recall computed in the experimental section ? what classifier is used here ?

---

> ### Author Response · Authors · 2023-11-20
> **Rebuttal Reviewer pzxF**
>
> We want to thank Reviewer pzxF for their feedback. We are glad that Reviewer pzxF found our paper to be “clearly presented”, with our theory bringing “clear answers” to the iterative retraining problem. We also appreciate that the reviewer found our experiments to be conclusive. We now address the specific comments raised by the reviewer below.
>
> We note that each of the reviewers questions has also been addressed in detail in the global response. Here we summarize these points but kindly ask the reviewer to also read our global response to all reviewers.
> - Assumption 3 implies sample complexity generalization bounds on the parameter for generative models. From a theoretical perspective, numerous works have proved sample complexity bound on the distribution, more precisely universal *density approximation* and *function approximation* capabilities of model classes like diffusion models, affine coupling and residual normalizing flows.
> - Precision and Recall  Intuitively, at the distribution level, precision and recall measure how the training and the generated distributions overlap. Precision can be seen as a measure of what proportion of the generated samples are contained in the “support” of the training distribution. On the other hand, recall measures what proportion of the training samples are contained in the “support” of the generated distribution.  If the training distribution and the generated distribution are the same, then precision and recall should be perfect. If there is no overlap—i.e. disjoint between the training distribution and the generated distribution, then one should have zero precision and zero recall More details and references are provided in the joint answer to reviewers and in Appendix G.2. Feel free to ask any clarification questions if needed.
>
> We hope that our responses here in conjunction with global response help answer the great questions raised by the reviewer. We politely encourage the reviewer to continue asking more questions or if possible consider a fresher evaluation of our paper with a potential score upgrade.

---

> ### Author Response · Authors · 2023-11-21
> **Kindly waiting for more feedback**
>
> We would like to thank you again for your positive and constructive comments. The global response should provide clarification on Assumption 3, precision, and recall. In addition, the updated experiments in Figure 3 now strengthen our contribution and corroborate better with our theory (Theorems 1 and 2). We appreciate your time and effort in this rebuttal period and hope our answers are detailed enough to convince you to upgrade your score.

---

### Author Response · Authors · 2023-11-20
**Global Response**

We would to thank the reviewers for all their detailed feedback which has allowed us to strengthen the updated manuscript. In particular, we are heartened to hear that all reviewers (pzxF, bfKK, 9HHH, LQsW) appreciated the clean presentation, writing, and structure of the paper. We are also glad that reviewers (pzxF, 9HHH) highlighted the experiments and overall technical rigor of our work. Lastly, we are encouraged that reviewers (LQsW, 9HHH, bfKKK) found our research question to tackle an interesting, important, and timely problem that is the “first work to study the theoretical stability of generative models when retrained on their own data”.

## Summary of PDF updates

All the main updates to the PDF are colored in blue. We now summarize the key updates of the paper before addressing the shared concerns raised by the reviewers.

- As asked by Reviewer 9HHH, we added more epochs and retraining for the experiments on real datasets. The graphs now show an even clearer convergence of the retrainings (see Figure 3).
- As asked by Reviewer LQsW, we added experiments of a flow method on CIFAR (see Figure 3). We used the recent conditional normalizing flow algorithm [1] [2]. The conclusions are the same for the flow method as for the diffusion-like methods, we observe the convergence of the iterative retraining for a sufficiently small amount of generated data (usually less than 50%).
- As asked by reviewer bfKK, we added quantitative synthetic experiments (2D experiments), showing the Wasserstein convergence of iterative retraining (see Appendix G.4).
- We added the visualization of EDM samples for FFHQ in Appendix G.3, which shows little to no degradation of the images for iterative retraining with a mix of real and synthetic data, as opposed to retraining exclusively on synthetic data.
- We have updated the proofs for single and multivariate Gaussian cases which now include additional Lemmas (see Lemma A.1 and A.2) that improve the structure, readability, and verifiability.


[1] Lipman, Y., Chen, R.T., Ben-Hamu, H., Nickel, M. and Le, M., 2022. Flow matching for generative modeling. arXiv preprint arXiv:2210.02747.

[2] Tong, A., Malkin, N., Huguet, G., Zhang, Y., Rector-Brooks, J., Fatras, K., Wolf, G. and Bengio, Y., 2023, July. Improving and generalizing flow-based generative models with minibatch optimal transport. In ICML Workshop on New Frontiers in Learning, Control, and Dynamical Systems.

---

> ### Author Response · Authors · 2023-11-20
> **Precision / Recall (Reviewer pzxF, 9HHH)**
>
> We thank the reviewer for their question regarding the exact computation of the Precision/Recall metric used for evaluation.
>
> We first highlight the notion of precision and recall used for generative models is different from the one used in standard supervised learning. It was first introduced in “Assessing Generative Models via Precision and Recall” [1] and refined in “Improved Precision and Recall Metric for Assessing Generative Models” [2], which we use in practice. Intuitively, at the distribution level, precision and recall measure how the training and the generated distributions overlap. Precision can be seen as a measure of what proportion of the generated samples are contained in the “support” of the training distribution. On the other hand, recall measures what proportion of the training samples are contained in the “support” of the generated distribution. If the training distribution and the generated distribution are the same, then precision and recall should be perfect. If there is no overlap—i.e. disjoint between the training distribution and the generated distribution, then one should have zero precision and zero recall.
>
> In order to adapt these metrics for empirical distributions that have finite samples customary of training deep generative models, multiple numerical tricks are required.  Specifically, for an empirical set of samples, the support of the associated distribution is approximated by taking the union of balls centered at each point with radius determined by the distance to the k-th nearest neighbor. This is generally done in some meaningful feature space (here we use the Inception network [3]). In order for our work to be self-contained, the remaining details on the precision and recall are now added in a new Appendix for additional clarity).
>
> ​​We appreciate the insightful feedback and excellent inquiries from the reviewer. Our rebuttal aims to comprehensively address the key issues highlighted. Should our answers meet your expectations, we would be grateful if you could reconsider and potentially upgrade your evaluation. Please do let us know.
>
> [1] M. S.-M. Sajjadi, O. Bachem, M. Lucic, O. Bousquet, and S. Gelly. Assessing generative models via precision and recall. Neurips 2018
>
> [2] Kynkäänniemi, Tuomas, et al. "Improved precision and recall metric for assessing generative models." Advances in Neural Information Processing Systems 32 (2019).
>
> [3] M. Heusel, H. Ramsauer, T. Unterthiner, B. Nessler, and S. Hochreiter.   Gans trained by a two-time-scale update rule converge to a local Nash equilibrium. NeurIPS 2017.

---

> ### Author Response · Authors · 2023-11-20
> **## How realistic is Assumption 3? (Reviewers pzxF, 9HHH)**
>
> We acknowledge the reviewer's healthy skepticism on the applicability of Assumption 3 in practice. Assumption 3 implies the existence of sample complexity bounds (on the parameters) on the considered generative models. From a theoretical perspective, there are numerous works that prove sample complexity bound on the distribution, more precisely universal *density approximation* capabilities of model classes like diffusion models (Oko et. al 2023), and affine coupling normalizing flows ([5], Teshima et. al 2020). Universal function approximation capabilities were also proven for residual flows ([6], Zhang et. al 2020). We would like to note that all of our experimental results use diffusion models or normalizing flows and thus from a theoretical perspective is in line with existing theory.
>
> From a practical point of view, our paper is motivated by the existence of web-scale models that are trained on the Internet. Examples of these models include Dalle-3, MidJourney [8], and StableDiffusion [9]. We argue that it is hard to debate the impressive power of these models, especially their ability to handle potentially never-before-seen prompts. That is the generalization ability of these models as evidenced by their sample quality reduces the concern that diffusion—and more generally likelihood-based generative models—given sufficient, data, and can capture the natural image manifold. Consequently, we believe that while Assumption 3 is needed for our theory, we can reasonably expect to satisfy it in practice when working with large foundation models that consume web-scale data.
>
>
> [4] K. Oko, S. Akiyama, and T. Suzuki. Diffusion models are minimax optimal distribution estimators. arXiv preprint arXiv:2303.01861, 2023.
>
> [5] T. Teshima, I. Ishikawa, K. Tojo, K. Oono, M. Ikeda, and M. Sugiyama. Coupling-based invertible neural networks are universal diffeomorphism approximators. Advances in Neural Information Processing Systems, 33:3362–3373, 2020.
>
> [6] Han Zhang, Xi Gao, Jacob Unterman, and Tom Arodz. Approximation capabilities of neural odes and invertible residual networks. In International Conference on Machine Learning, pp. 11086–11095. PMLR, 2020.
>
> [7] A. Ramesh, M. Pavlov, G. Goh, S. Gray, C. Voss, A. Radford, M. Chen, and I. Sutskever. Zero-shot text-to-image generation. In International Conference on Machine Learning. PMLR, 2021.
>
> [8] Midjourney. https://www.midjourney.com/home/, 2023. Accessed: 2023-09-09.
>
> [9] Stability AI. https://stability.ai/stablediffusion, 2023. Accessed: 2023-09-09.

---

> ### Author Response · Authors · 2023-11-20
> **## Improving Empirical Evaluation**
>
> We acknowledge the reviewers's comments regarding the number of seeds used in our iterative retraining experiments for natural image datasets. We first highlight that we have improved this experiment in the updated version of the manuscript but substantially increased the number of iterative retraining steps ($5$ vs $20$) while adding a fully synthetic baseline ($\lambda = \infty$). In our plots, we see that the margin of difference between the fully synthetic case and the more stable regime of $\lambda > 0$ of iterative retraining is quite dramatic. While the reviewer has requested more random seeds, we wish to politely push back due to the high computational cost of running each experiment. For example, just for FFHQ-64, conducting iterative retraining using the SOTA EDM model amounts to a full week of experimentation using 40 RTX8000 GPUs. Unfortunately, running additional random seeds for all models and datasets is beyond the computational budget within the period of this rebuttal. However, to promote reproducible research we commit to releasing the full training code of all our experiments such that one can readily verify the stability and collapse property of iterative retraining. In addition, as required by reviewer bfKK, we have added quantitative measurements for the experiments on the 2D synthetic data and were able to repeat this experience multiple times and provide statistics such as confidence intervals and error bars.
>
> **Empirical evaluation of toy models**
>
> We thank the reviewer for their great suggestion. We have now included a plot of the Wasserstein 1 and Wasserstein 2 distance between the generated samples and the ground truth dataset as a function of the number of retraining steps (Figure 6 in Appendix G.4). These quantitative results add further support to the observed visual phenomena of collapse and stability highlighted in Fig 1 and 2.

---

### Meta-Review · Area_Chair_HgBM · 2023-12-13

**Metareview:**

This is an interesting and timely paper which provides a theoretical understanding of when generative models iteratively re-trained on their own data diverge from the original target distribution. While reviewers had some reservations regarding the validity of the assumptions made in the analysis, overall these theoretical results, and their empirical validation, will likely be of broad interest to the community.

**Justification For Why Not Higher Score:**

This is a nice paper, but I don't think it lends itself terribly well to an oral. The results are mostly theoretical and could be summarized succinctly.

**Justification For Why Not Lower Score:**

I think this paper will be of broad interest. It also is going to be somewhat contentious (e.g. one reviewer remains skeptical even after agreeing the theory is technically sound, given the assumptions; other related papers have appeared on arXiv investigating the same problem as well). A spotlight would be nice. However it is not essential.

---

### Decision · Program_Chairs · 2024-01-16

Accept (spotlight)